# Blue-to-Red TagFT, mTagFT, mTsFT, and Green-to-FarRed mNeptusFT2 Proteins, Genetically Encoded True and Tandem Fluorescent Timers

**DOI:** 10.3390/ijms24043279

**Published:** 2023-02-07

**Authors:** Oksana M. Subach, Anna V. Vlaskina, Yulia K. Agapova, Alena Y. Nikolaeva, Konstantin V. Anokhin, Kiryl D. Piatkevich, Maxim V. Patrushev, Konstantin M. Boyko, Fedor V. Subach

**Affiliations:** 1Complex of NBICS Technologies, National Research Center “Kurchatov Institute”, Moscow 123182, Russia; 2Bach Institute of Biochemistry, Research Centre of Biotechnology of the Russian Academy of Sciences, Moscow 119071, Russia; 3Laboratory for Neurobiology of Memory, P.K. Anokhin Research Institute of Normal Physiology, Moscow 125315, Russia; 4Institute for Advanced Brain Studies, M.V. Lomonosov Moscow State University, Moscow 119991, Russia; 5Westlake Laboratory of Life Sciences and Biomedicine, Hangzhou 310024, China; 6Institute of Basic Medical Sciences, Westlake Institute for Advanced Study, Hangzhou 310024, China

**Keywords:** genetically encoded blue-to-red fluorescent timers, protein engineering, fluorescence imaging, fluorescent protein, mTagFT, TagFT, mTsFT, mNeptusFT2, FucciFT2, crystal structure

## Abstract

True genetically encoded monomeric fluorescent timers (tFTs) change their fluorescent color as a result of the complete transition of the blue form into the red form over time. Tandem FTs (tdFTs) change their color as a consequence of the fast and slow independent maturation of two forms with different colors. However, tFTs are limited to derivatives of the mCherry and mRuby red fluorescent proteins and have low brightness and photostability. The number of tdFTs is also limited, and there are no blue-to-red or green-to-far-red tdFTs. tFTs and tdFTs have not previously been directly compared. Here, we engineered novel blue-to-red tFTs, called TagFT and mTagFT, which were derived from the TagRFP protein. The main spectral and timing characteristics of the TagFT and mTagFT timers were determined in vitro. The brightnesses and photoconversions of the TagFT and mTagFT tFTs were characterized in live mammalian cells. The engineered split version of the TagFT timer matured in mammalian cells at 37 °C and allowed the detection of interactions between two proteins. The TagFT timer under the control of the minimal arc promoter, successfully visualized immediate-early gene induction in neuronal cultures. We also developed and optimized green-to-far-red and blue-to-red tdFTs, named mNeptusFT and mTsFT, which were based on mNeptune-sfGFP and mTagBFP2-mScarlet fusion proteins, respectively. We developed the FucciFT2 system based on the TagFT-hCdt1-100/mNeptusFT2-hGeminin combination, which could visualize the transitions between the G1 and S/G2/M phases of the cell cycle with better resolution than the conventional Fucci system because of the fluorescent color changes of the timers over time in different phases of the cell cycle. Finally, we determined the X-ray crystal structure of the mTagFT timer and analyzed it using directed mutagenesis.

## 1. Introduction

Genetically encoded FTs change their fluorescent color over time. FTs have been used to monitor the activation and downregulation of promoters [1], determine the protein cellular trafficking pathway [2], label engram neuronal populations in the mouse brain involved in two episodes of learning [3], estimate changes in protein synthesis and degradation rates [4], determine the secretory vesicle age [5], and differentiate primary infected neurons from secondary neurons infected with rabies virus [6]. FTs can be divided into three different classes according to the composition and chemical transformations of their chromophores.

Tandem fluorescent timers (tdFTs) are composed of two fluorescent proteins of different fluorescence colors, which have fast and slow maturation; these two proteins are linked together via a linker within one molecule. A number of tdFTs have been engineered, such as green/large Stokes shift orange sfGFP-LSSmOrange [7], green/orange sfGFP-mOrange2 [4], green/red sfGFP-mCherry [8,9], EGFP-Tdimer2 [10], mKate-sfGFP, TagRFP-sfGFP, and tdTomato-sfGFP [11]. However, blue-to-red and green-to-far-red tdFTs have not been developed. There is a need for additional tdFTs, since green-to-far-red tdFTs, spectrally compatible with the available blue-to-red fluorescent timers, would allow multiplexing imaging, and the blue-to-red tdFTs could address the low brightness of the red form of the available blue-to-red timers.

Pseudo fluorescent timers (psFTs) encode one polypeptide molecule and two forms of chromophores of different colors, that mature independently with fast and slow rates. The DsRed-E5 green-to-red tetrameric timer first exhibits green fluorescence; over time, red fluorescence appears, and the green fluorescence becomes dimmer as a consequence of Förster resonance energy transfer (FRET) from the green to the red subunits in the tetramer [1,12]; the green and red forms are dead end products [12]. The green-to-orange mK-GO timer could also be classified as a pseudo timer since both forms are dead end products [5].

One fluorescent form of a true fluorescent timer (tFT) transforms into a fluorescent form of a different color over time. This class of tFTs includes the mCherry-derived blue-to-red FTs: FastFT, MediumFT, and SlowFT [2], and the mRuby-derived blue-to-red FT, mRubyFT [13]. These tFTs are monomeric and allow the labeling of individual proteins. mRubyFT is currently the best blue-to-red FT, since it has an increased brightness of the blue form and a chemically stable red form, and is less prone to blue-to-red photoconversion [13]. However, the applicability of the mRubyFT timer for different biological applications has not been shown, and the brightness of the red form is still 13-fold lower compared to the standard EGFP protein. Also, tdFTs and tFTs have not been compared with each other under the same experimental conditions.

In this article, we developed true blue-to-red fluorescent timers based on the TagRFP protein and blue-to-red and green-to-far-red tandem timers, characterized and compared their properties in vitro and in mammalian cells, and applied these timers to different biological applications. A split version of the splitTagFT timer allowed the detection of interactions between proteins in mammalian cells. The TagFT timer, under the control of the minimal arc promoter, successfully visualized the induction of immediate-early genes in neuronal cultures. Based on this combination of the true and tandem timers, we developed the FucciFT2 system to accurately visualize transitions between phases of the cell cycle. Finally, we resolved the crystal structure of the mTagFT protein.

## 2. Results and Discussion

### 2.1. Engineering Blue-to-Red Fluorescent Timers Based on TagRFP in E. coli

To develop the genetically encoded blue-to-red fluorescent timer, the TagRFP protein was subjected to directed and random mutagenesis followed by screening and selection of bacterial libraries using an arabinose-inducible system. In the first step, we subjected the red fluorescent protein (RFP) TagRFP to directed saturated mutagenesis at positions 18, 69, 83, 84, 148, 152, 165, 179, 181, 203, 205, and 224 (Figure 1), using combination mutagenesis at different sites simultaneously. However, we could not find any clone with a blue-to-red timer phenotype. We next applied TagRFP to the nine rounds of random mutagenesis, followed by the screening procedure as described previously in [13]. Briefly, we selected the bluest/nonred and nonblue/reddest colonies on Petri dishes supplied with 0.2% arabinose at 18 and 72 h after library plating and incubation at 37 degrees. As a result, we found a final variant named TagFT (Figure 1 and Appendix A).

To characterize the oligomeric properties of the timers in vitro on purified proteins, we ran the proteins on a seminative polyacrylamide gel (PAAG, Appendix A), or performed fast protein liquid chromatography (FPLC, Figure 2e). TagFT was run on a 0.5% sodium dodecyl sulfate (SDS) gel as a monomer (Appendix A). However, the TagFT timer on FPLC eluted mostly as a dimer with an admixture of tetramers and oligomers. The presence of 0.5% SDS was most likely sufficient to destroy the weak dimeric interface of the TagFT protein. According to the alignment of amino acid sequences (Figure 1), the TagFT timer contains six external mutations that could lead to protein oligomerization.

To monomerize the dimeric TagFT timer, external mutations that appeared during random mutagenesis were removed by overlap mutagenesis, and site-directed mutagenesis was performed at internal positions 64, 83, 165, 181, 220, and 224, which are probably the key to the timer phenotype (Figure 1). We additionally introduced the external substitutions V151K, Y153K, and C229S to further disrupt the dimerization interfaces, as described for the PATagRFP protein [14]. As a result of the site-directed mutagenesis followed by screening on Petri dishes, as described above, we found clones with a blue-to-red timer phenotype, i.e., in which the blue form turned red over time. Several of the brightest mutants were used as templates for a subsequent round of random mutagenesis followed by screening, as described above. One round of random mutagenesis did not lead to a noticeable improvement in the brightness of both forms, so we decided that we had reached saturation by these criteria and stopped optimizing the monomeric timer, and the brightest variant was chosen and named mTagFT (Figure 1 and Appendix A). Indeed, PATagRFP’s monomerizing mutations worked for the TagFT timer, since mTagFT ran on 0.5% SDS PAAG as a monomer (Appendix A), and on FPLC, it eluted primarily as a monomer with an admixture of oligomers (Appendix A). Since we observed the formation of the disulfide bond between the C118 residues from different subunits in the X-ray structure of the mTagFT protein (see data below), we introduced the C118S mutation to prevent these intersubunit interactions and reduce the formation of the oligomers. However, the mTagFT/C188S mutant had a similar percentage of oligomers on FPLC (Appendix A).

### 2.2. Characterization of the Purified TagFT and mTagFT Timers In Vitro

First, we assessed the spectral and biochemical properties of the TagFT and mTagFT proteins isolated from bacterial cells and compared them to the properties of mRubyFT (Figure 2 and Appendix A, and Table 1).

The blue forms of the TagFT and mTagFT timers had absorption/excitation/emission maxima at 405/408/457 and 404/406/461 nm, respectively (Figure 2 and Appendix A, and Table 1). The red forms of the TagFT and mTagFT timers had absorption/excitation/emission maxima at 557/556/582 and 562/557/590 nm, respectively (Figure 2 and Appendix A, and Table 1). Compared to the mRubyFT timer, the absorption/excitation/emission for the blue forms of the TagFT and mTagFT timers were similar, and those of the red forms were blueshifted by 15-20/25-26/34-42 nm, respectively. Hence, the spectral properties of the TagFT, mTagRF, and mRubyFT timers were practically the same, except that the excitation and emission maxima for the red forms of mTagFT and TagFT were blueshifted compared to those of mRubyFT.

The molecular brightnesses of the blue forms of the TagFT and mTagFT timers, determined by acid denaturation, were 1.5-fold lower and 1.3-fold larger, respectively, than the brightness of the corresponding form of the mRubyFT protein (Table 1). The molecular brightnesses of the red forms of the TagFT and mTagFT timers, determined by the alkaline denaturation method, were 13-fold and 1.7-fold greater, respectively, than the brightness of the corresponding form of the mRubyFT protein (Table 1). The brightnesses of the blue and red forms of TagFT and mTagFT, determined relative to the absorption at 280 nm, were similar to and 1.2–1.8-fold lower than, the corresponding forms for the mRubyFT protein, respectively (Table 1). The discrepancy in extinction coefficient values determined by the denaturation method relative to the absorption at 280 nm could be attributed to the inefficient folding of the proteins in bacterial cells. The brightness of the red forms relative to absorption at 280 nm resembled the brightness of the red forms determined in mammalian cells (please see Section 2.3 below). Hence, the brightnesses of the TagFT and mTagFT timers purified from bacterial cells were similar to, or 1.2–1.8-fold lower than, that of the mRubyFT protein, respectively.

To validate the timer characteristics of the TagFT and mTagFT timers, we expressed these proteins in the restriction of oxygen conditions, purified them at low temperature (0–4 °C), and registered their maturation at 37 °C in a cuvette. The blue fluorescence of the TagFT and mTagFT proteins reached maxima at 0.83 and 4.7 h, and completely disappeared by 12 and 50 h, respectively (Figure 2c and Appendix A, and Table 1). The maturation half-times for the red form of the TagFT and mTagFT timers were observed at 2.7 and 8 h, respectively (Figure 2c and Appendix A and Table 1). According to the characteristic time for the red form, TagFT and mTagFT matured 5.5-fold and 1.9-fold faster than mRubyFT, respectively (Table 1). The red form of TagFT matured 1.4-fold faster than mCherry-derived MediumFT [2]. The red form of mTagFT matured at a rate similar to the FastFT and SlowFT timers. The blue-to-red ratio values for TagFT and mTagFT reached values of 40 in 8 and 40 h, respectively (Figure 2c and Appendix A). Hence, TagFT displayed the fastest maturation rate of the red form among blue-to-red timers, and mTagFT matured similarly to the FastFT and SlowFT timers.

We then evaluated the pH stabilities of the TagFT and mTagFT timers (Figure 2d and Appendix A, and Table 1). The sensitivities of the blue and red fluorescences of the TagFT and mTagFT timers to variations in pH were similar, with pKa values of 4.8–5.32, except for the unusually strong pH sensitivity of the blue form, with a pKa value of 7.3 for the TagFT timer (Figure 2d and Appendix A, and Table 1). Hence, the fluorescences of the TagFT and mTagFT timers varied in the physiological pH range of 4.5–8.0, especially in the case of the blue fluorescence of the TagFT timer. This pH-sensitivity may be problematic in the physiological pH range, where timing effects could be obscured by pH changes, leading to problems in quantification.

To estimate the oligomeric states of the TagFT and mTagFT purified timers in vitro, we loaded proteins onto heminative polyacrylamide gel electrophoresis (PAGE, Appendix A) and characterized the elution times of the proteins using fast protein liquid chromatography (FPLC, Figure 2e and Appendix A). TagFT and mTagFT ran as monomers on heminative PAGE supplemented with 0.5% sodium dodecyl sulfate (SDS, Appendix A). On FPLC, TagFT and mTagFT eluted as a dimer and monomer, respectively, with a noticeable admixture of oligomers (Figure 2e and Appendix A). The discrepancy between the oligomeric states of the TagFT protein determined by the two methods can be explained by the fact that 0.5% SDS denaturant was sufficient to disrupt the dimeric interface of the TagFT protein. Hence, TagFT and mTagFT existed as a dimer and a monomer in water solution, respectively.

For the first time, we compared the photostabilities of the red forms of the TagFT and mTagFT proteins with that of the FastFT timer (Figure 2f and Appendix A). Under continuous 550/25 nm light illumination (from a mercury lamp) of the protein aqua drops in oil, through a 63× oil objective lens, the red fluorescences of the TagFT, mTagFT, and FastFT timers photobleached to 50% at 605 ± 120, 1221 ± 282, and 560 ± 69 s, respectively. The TagFT and mTagFT photobleaching behavior looked similar to the photobleaching curve of TagRFP-T, and reflected photochromism, likely involving cis-trans isomerization, and bleaching happening simultaneously [17]. We speculate that an A165S mutation could be responsible for the two-fold higher photostability of the mTagFT timer as compared to TagFT, since mutation in the TagRFP ancestor protein at the same position resulted in a nine-fold greater photostability [17]. Hence, the red forms of the TagFT and mTagFT timers revealed photochromism, and demonstrated similar and 2.2-fold higher photostabilities, compared to the photostability of the FastFT timer.

### 2.3. Behavior of the TagFT and mTagFT Timers in Cultured Mammalian Cells

We next assessed the brightnesses and blue-to-red photoconversions of the TagFT and mTagFT timers in mammalian cells, and compared them with the corresponding characteristics of the control mRubyFT true timer and mTsFT tandem timer (see Section 2.7 for the development of the mTsFT).

To characterize the brightnesses of the TagFT, mTagFT, mTsFT, and control mRubyFT timers, they were transiently expressed in HeLa mammalian cells, and the brightnesses of the blue and red forms were characterized 24 and 72 h after transfection, respectively. The brightnesses of both forms were normalized to the brightness of the EGFP coexpressed in the same cells via the P2A self-cleavable peptide in the FT-P2A-EGFP fusion. The blue and red fluorescences of TagFT, mTagFT, and mTsFT were detected 24–72 h after transfection, and all timers were evenly distributed across the HeLa and HEK293T cells (Figure 3a,f, Appendix A). The normalized brightness values of the blue forms of TagFT, mTagFT, and mTsFT were similar, and 1.39-fold and 2.6-fold higher than that of the control mRubyFT timer, respectively (Figure 3b and Table 2). The normalized brightnesses of the red forms of the TagFT, mTsFT, and mTagFT timers were 1.63-fold and 24-fold higher, and 1.31-fold lower, than that of the mRubyFT protein, respectively (Figure 3c and Table 2). Hence, compared to mRubyFT, the brightnesses of the TagFT and mTagFT true FTs were the same or differed by up to 1.63-fold, but the brightness of the mTsFT tandem timer was 2.6–24-fold larger; hence, the tandem blue-to-red mTsFT timer was superior to true blue-to-red timers in terms of brightness.

The mRubyFT timer in mammalian cells has been shown to be prone to blue-to-red photoconversion under 395 nm light, but to a lesser extent than FastFT [13]. We compared the susceptibilities of the TagFT, mTagFT, mTsFT, and control mRubyFT timers to the light-induced blue-to-red transition when expressed in HEK293T cells (Figure 3d–f, Appendix A). The timers were transiently expressed in HEK293T cells, and 24 h after transfection, the cells were continuously illuminated with violet light (395/25 nm, 0.338 mW/cm^2^ before objective lens) or cyan light (433/25 nm, 0.920 mW/cm^2^ before objective lens) for 1 min. The respective values were normalized to the light power for accurate comparison.

Illumination with violet light (395/25 nm) resulted in a significant decrease in the blue fluorescences and a significant increase in the red fluorescences of the TagFT, mTagFT, and control mRubyFT timers with ΔF/F values varying in the ranges of −3.2–−1.8 and 0.65–5.1, respectively (Figure 3d–f and Appendix A). The blue and red fluorescence emissions of the tandem mTsFT timer were not affected by exposure to violet light, since the ΔF/F values varied in the range of −0.12–−0.05 (Figure 3d and Appendix A). Compared to mRubyFT, the efficiencies of blue-to-red photoconversion with 395/25 nm light for TagFT and mTagFT were similar or 5-fold larger, respectively (Figure 3e).

Illumination with cyan 433/25 nm light also resulted in a decrease in the blue fluorescences and an increase in the red fluorescences of the TagFT, mTagFT, and control mRubyFT timers, with ΔF/F values in the range of −0.46–−0.28 and 0.26–2.4, respectively (Appendix A). The fluorescence of the tandem mTsFT timer was not affected by exposure to cyan light since the ΔF/F values were in the range of −0.008–0.002 (Appendix A). Compared to mRubyFT, the efficiencies of the blue-to-red photoconversion with 433/25 nm light for TagFT and mTagFT were the same or 15-fold larger, respectively (Appendix A). Compared to violet light, cyan light affected the blue and red fluorescences of the TagFT, mTagFT, and mRubyFT timers to a 3.9–9.9-fold and 2.2–2.4-fold lesser extent, respectively (Appendix A). Compared to violet light, the efficiencies of blue-to-red photoconversion with cyan light were 1.69-fold lower, the same, and 1.73-fold higher for mRubyFT, TagFT, and mTagFT, respectively (Appendix A). Since the absorption maxima of the blue forms of the mRubyFT, TagFT, and mTagFT timers were practically the same (peaked at approximately 404–406 nm), and we did not observe other absorption peaks, we explained the differences in the efficiencies of the blue-to-red phototransformation under cyan light by the presence of different oligomeric states and the diverse close surroundings of the chromophores of the timers. Overall, compared to mRubyFT, we revealed similar or 5–15-fold higher blue-to-red photoconversions of the TagFT and mTagFT timers, respectively, with both violet or cyan lights in mammalian cells; since we did not observe blue-to-red photoconversion in mammalian cells in the case of the tandem mTsFT timer, it was preferable in this regard.

### 2.4. Visualization of the mTagFT Timer in Fusions with Cytoskeletal Proteins in Mammalian Cells

Since mTagFT was monomeric, we next studied its behavior in fusions with cytoskeleton proteins in mammalian cells. At 72 h after transfection of HeLa cells, we could image both the blue and red forms of mTagFT-β-actin and mTagFT-α-tubulin, but the fluorescence was dim (Figure 4a,b). The calculated red-to-blue ratios varied within the cell for all fusions tested, similar to analogous fusions of the mRubyFT timer [13]. These images would argue against the monomeric nature of the mTagFT timer in mammalian cells and were likely related to the presence of oligomers admixed in vitro (Appendix A). Less difficult fusions of mTagFT (Figure 4c) and mTsFT, mNeptusFT2 (see Section 2.7) with vimentin protein, were bright and properly localized in HeLa cells. However, TagFT demonstrated uneven brightness of the vimentin fibers, which was probably related to its dimeric state (Figure 4d). So monomerization of the TagFT timer assisted correct filament formation in the case of vimentin. Hence, mTagFT is not recommended for labeling β-actin and α-tubulin due to its low brightness, but in contrast to dimeric TagFT can be used for the targeting of vimentin.

### 2.5. Characterization of the Split Versions of Blue-to-Red Timers in Mammalian Cells

The development of split versions of the blue-to-red timers would allow researchers to visualize the interaction between two proteins and judge the time elapsed since the start of the event. We made split versions of the mRubyFT, TagFT, mTagFT, and MediumFT timers between residues D158 and G159 (Figure 1), by analogy to split versions of the RFPs mKate (between residues 151-152), mCherry (between residues 159–160), and mLumin (between residues 151–152), maturing at 37 °C [18]. We fused the N- and C-terminal split parts of timers to bJun (bJun-FTN) and bFos (bFos-FTC) heterodimerizing proteins (Figure 5a), and bFosΔZip (bFosΔZip-FTC) as a control, respectively, as described previously [19]. HeLa cells transfected with the bJun-FTN, bFos-FTC, or bFosΔZip-FTC fusions alone were nonfluorescent after 24 h of incubation at 37 °C.

Coexpression of the generated bJun-FTN and bFos-FTC split versions in HeLa mammalian cells at 37 °C for 24 h revealed the absence of both blue and red fluorescence, except for the TagFT and mRubyFT timers, which matured at 37 °C and showed blue and red fluorescence in the nuclei of the cells (Figure 5b). Compared to mRubyFT, the blue and red fluorescences of the TagFT timer were 3.2-fold and 4.0-fold brighter, respectively (Figure 5c). Incubation of the cells at a lower temperature of 25 °C for an additional 24 h resulted in the appearance of both blue and red fluorescence for the mTagFT and MediumFT timers. Hence, among the mRubyFT, mTagFT, TagFT, and MediumFT timers, the split version of the TagFT timer matured more efficiently in mammalian cells at 37 °C.

Compared to the heterodimerizing bJun-TagFTN/bFos-TagFTC pair, the blue and red fluorescences of the cells coexpressing the noninteracting control proteins bJun-TagFTN and bFosΔZip-TagFTC were dimmer by 1.8-fold and 2.3-fold, respectively (Figure 5d). Hence, split TagFT allows the detection of the interaction between two proteins with ΔF/F values of 80–130%.

In summary, the split version of the TagFT timer matured in mammalian cells at 37 °C and could detect interactions between two proteins.

### 2.6. Application of the TagFT Blue-to-Red Fluorescent Timer under the Control of the Arc Promoter for Visualization of the Activation of IEG in Neurons

The promoters c-fos and arc are immediate-early gene promoters, and their activation results in long-term synaptic plasticity and memory formation [20]. The Slow-FT timer under the control of the TRE promoter in the c-fos:tTA/TRE system was used for the in vivo labeling of engram cells, which are activated in two episodes of learning separated by time [3]. To apply the TagFT timer for the marking of such activated engram neuronal populations, we cloned the TagFT timer under the control of the minimal arc promoter [21], and visualized the activation of the arc promoter in neuronal cells.

For this experiment, we chose the TagFT timer because it demonstrated the largest brightness in mammalian cells among other true timers (Table 2). Tandem timers were not optimal for this application, since the faster maturing forms did not disappear over time. The TagFT timer was cloned into the pAAV-Arc1-TagFT-3xNLS vector. The minimal size of the arc promoter (555 bp) was chosen to achieve efficient packaging of the rAAV particles (Appendix A). To facilitate cell counting, the nuclear localization signal (3xNLS) was added to the C-terminus of the TagFT timer (Appendix A). The expression of the blue and red forms of the TagFT timer under the control of the arc1 promoter was tested in HEK293T mammalian cells. Twenty-four hours after transfection, we observed both bright blue and blue/red cells with nuclear localization (Appendix A).

Next, the pAAV-Arc1-TagFT-3xNLS construct was tested in a neuronal culture obtained from the hippocampus of transgenic Barth mice (B6.Cg-Tg(Fos/EGFP)1-3Brth/J line) [22], which expresses EGFP as a result of the activation of the c-fos early gene promoter. To infect a neuronal culture, recombinant AAV virus particles encoding the *Arc1-TagFT-3xNLS* gene were purified and added to the cell culture on the fourth day in vitro (DIV). Two to three weeks after viral transduction of the neuronal cultures, chemical induction of early gene expression was performed, followed by imaging on a confocal microscope. After chemical induction, we observed expression of the EGFP protein and the TagFT timer from the c-fos and arc1 promoters in different and in the same cells, with maximal fluorescence of the EGFP and blue-form of the TagFT timer at 18 ± 11 h (two cultures, four cells) and 25 ± 5 h (two cultures, four cells), respectively. As an example, at 11 and 20 h after chemical induction with potassium chloride, we observed green and blue fluorescence maxima in the nucleus of the same neuron, corresponding to the maximal activity of the c-fos and arc1 promoters, respectively (Figure 6). At later times, the green fluorescence of the c-fos promoter disappeared, and the blue form of the TagFT timer under the control of the arc1 promoter, turned red (Figure 6). Hence, the TagFT timer can be used for detecting immediate-early gene induction in neuronal cultures, and could potentially be applied to mark the activity of engram neuronal populations involved in two temporally separated episodes of learning in vivo.

### 2.7. Use of Fluorescent Timers for Cell Cycle Visualization in Mammalian Cells

The Fucci and Fucci4 systems were developed to visualize the transition between G1 and S/G2/M phases of the cell cycle using the cell cycle-dependent degradation of permanently fluorescent proteins [23,24]. We suggest that the replacement of the permanently fluorescent proteins in the Fucci system with fluorescent timers, should provide additional information about the longevity of the G1 and S/G2/M phases. However, the available blue-to-red FTs would allow labeling of only one phase of the cell cycle, and there is a need to engineer FTs with compatible spectrally resolved forms, such as green and far-red forms.

To address the need for green-to-far-red FTs, we engineered tandem FTs based on the fusion of superfolder GFP (sfGFP) and variants of the mNeptune far-red FP with slow maturation. To slow the maturation rate of the mNeptune protein in the mNeptune-sfGFP fusion, we performed one round of random mutagenesis of the mNeptune protein followed by screening of slowly maturing mutants with far-red fluorescence on Petri dishes, as described above. As a result, the mNeptusFT1 and mNeptusFT2 tandem timers were chosen (the name follows mNeptune-sfGFP fusion based Fluorescent Timer). mNeptusFT1 had the same amino acid sequence as the original mNeptune, and mNeptusFT2 contained one external mutation, D159N (Appendix A). The spectral and biochemical properties of the mNeptusFT1 and mNeptusFT2 timers were characterized in vitro, and mNeptusFT2 properly localized when expressed in fusion with vimentin in HeLa cells, which correlated with its monomeric state in vitro (Table 1, Appendix A). Since mNeptusFT1 does not have mutations, and mNeptusFT2 has only one external mutation, we suggest that they should have the same photostabilities as the mNeptune and sfGFP proteins [15,25].

To improve the brightness of the blue-to-red FTs, we also developed tandem blue-to-red FTs based on fusion of the bright mTagBFP2 BFP [26] and bright mScarlet RFP proteins [16]. Since the blue fluorescence in the mTagBFP2-mScarlet fusion was dim, in contrast to the bright red fluorescence, we subjected the mTagBFP2 part to random mutagenesis and selected the clones on Petri dishes with the brightest blue fluorescence, as described above. Finally, we found a mutant called mTsFT (the name follows mTagBFP2-mScarlet fusion based Fluorescent Timer), which contained one mutation in the mTagBFP2 region (Appendix A). The K173R mutation was external to β-can and probably facilitated the folding of the TagBFP2 part in fusion with the mScarlet protein. The spectral and biochemical properties of the mTsFT timer were characterized in vitro and it properly localized when expressed in fusion with vimentin in HeLa cells, which correlated with its monomeric state in vitro (Table 1 and Appendix A). Since mTsFT has only one external mutation, we suggest that it should have the same photostability as the mTagBFP2 and mScarlet proteins [16,26].

We next fused the blue-to-red true fluorescent timers mTagFT, mRubyFT, MediumFT, TagFT, FastFT, and FastFT2 with hCdt_1−100_, a protein accumulated in the nucleus of the cell in G1 phase according to a previous publication [24], and compared their brightness in HeLa mammalian cells at stable expression (Figure 7a). All fusions showed nuclear localization in the G1 phase (Figure 7a). The brightnesses of the blue forms were similar for all fusions of timers tested, except for a 1.7-fold lower brightness of FastFT-hCdt_1–100_ and practically absent blue fluorescence of the FastFT2-hCdt_1–100_ fusion (Figure 7a). The FastFT2 protein corresponded to the Blue124/I146S mutant, which demonstrated better characteristics than the FastFT timer [27]. The MediumFT-hCdt_1-100_ fusion demonstrated 10-fold and 4-fold brighter red fluorescence than the mTagFT-hCdt_1-100_ and mRubyFT-hCdt_1–100_ fusions, respectively, and a similar brightness to the TagFT-hCdt_1–100_ protein (Figure 7a).

Next, we compared the brightnesses of the best true blue-to-red MediumFT and TagFT timers, with tandem blue-to-red mTsFT timers in their fusions with hCdt_1–100_ protein at stable expression in HeLa cells (Figure 7b); we also engineered and stably expressed mNeptusFT2-hCdt_1–100_ fusion in HeLa cells. The blue form of the mTsFT-hCdt_1–100_ fusion was 1.84-fold and 2.3-fold brighter than the TagFT-hCdt_1–100_ and MediumFT-hCdt_1–100_ fusions (Figure 7b). The red form of the mTsFT-hCdt_1–100_ fusion was 15-fold brighter than both the TagFT-hCdt_1–100_ and MediumFT-hCdt_1–100_ fusions (Figure 7b). All timers in fusion with hCdt_1–100_ localized in the nuclei of the cells (Figure 7b). Cells expressing the mTsFT-hCdt_1-100_ fusion grew slowly and had unusual morphology, in contrast to the cells expressing TagFT-hCdt_1–100_ and MediumFT-hCdt_1–100_ fusions. When stably expressed in HeLa cells, the mNeptusFT2-hCdt_1–100_ fusion did not degrade in S/G2/M phases. Hence, the mTsFT, MediumFT and TagFT fusions with hCdt_1–100_ were chosen for further visualization of the G1 phase of the cell cycle.

We further compared the brightnesses of the blue-to-red TagFT and MediumFT true timers with the tandem mTsFT timer in fusion with the hGeminin protein, which accumulated in the S/G2/M phase in the nucleus of the cells, as described previously [24]. The blue fluorescence of the mTsFT-hGeminin fusion stably expressed in HeLa cells was 1.77-fold and 1.55-fold brighter than those of the TagFT-hGeminin and MediumFT-hGeminin fusions, respectively (Figure 7c). The red form of the mTsFT-hGeminin fusion was 20-fold and 9-fold brighter than the red forms of the TagFT-hGeminin and MediumFT-hGeminin fusions, respectively (Figure 7c). The mTsFT-hGeminin fusion accumulated in the S/G2/M phase in the nucleus (Figure 7c) and preserved normal cell division. Hence, the blue-to-red mTsFT-hGeminin fusion was preferable for visualization of the S/G2/M phase of the cell cycle compared to the blue-to-red TagFT-hGeminin and MediumFT-hGeminin fusions in terms of brightness.

For visualization of the G1 phase, we also assembled green-to-far-red mNeptusFT1-hCdt_1–100_ and mNeptusFT2-hCdt_1-100_ fusions and compared their brightnesses in HeLa cells (Figure 7d). Both mNeptusFT1-hCdt_1-100_ and mNeptusFT2-hCdt_1–100_ fusions, when stably expressed in HeLa cells, accumulated in the nuclei of the cells in G1 phase and did not hinder cell division (Figure 7d). The mNeptusFT1-hCdt_1–100_ and mNeptusFT2-hCdt_1-100_ fusions had similar brightnesses to the green form, and the far-red form of the mNeptusFT2-hCdt_1–100_ fusion was 1.62-fold brighter (Figure 7d). Hence, the green-to-far-red mNeptusFT2-hCdt_1–100_ fusion was preferable for visualization of the S/G2/M phase.

Finally, we visualized the transition between the G1 and S/G2/M phases of the cell cycle using the blue-to-red tandem mTsFT-hCdt_1–100_ and true TagFT-hCdt_1–100_ and MediumFT-hCdt_1–100_ fusions in combination with green-to-far-red mNeptusFT2-hGeminin fusions, the system called FucciFT (Figure 7e). We selected several (three for each of the mTsFT, TagFT and MediumFT timers) stable HeLa cell lines with the fastest growth, which expressed the blue-to-red FT-hCdt_1–100_ fusion together with the green-to-far-red mNeptusFT2-hGeminin fusion.

The blue/red fluorescence of the MediumFT-hCdt_1–100_/mNeptusFT2-hGeminin combination was either absent in two lines or dim in one line, so we did not choose this combination.

The lines with the mTsFT-hCdt_1-100_/mNeptusFT2-hGeminin combination, called FucciFT1, had bright blue/red fluorescence. Green fluorescence appeared at the beginning of the S/G2/M phases, became green/far-red at the end of these phases, and disappeared after the transition into G1 phase; and blue fluorescence appeared at the beginning of the G1 phase, became blue/red at the end of this phase (Appendix A), and disappeared after the start of the S/G2/M phases. We noted that the cells expressing the FucciFT1 system grew slowly, and the cell cycle was elongated (G1 phase was longer than 95 h).

The three selected stable lines with the TagFT-hCdt_1-100_/mNeptusFT2-hGeminin combination, called FucciFT2, were all blue/red fluorescent and grew quickly. Blue fluorescence appeared at the beginning of the G1 phase, became blue/red at the end of this phase, and disappeared after the transition into S phase; and green fluorescence appeared at the beginning of the S/G2/M phases, became green/far-red at the end of these phases, and disappeared after the start of the S/G2/M phases. (Figure 7e,f and Appendix A). The longevities of the G1 and S/G2/M phases of the FucciFT2 expressing HeLa cells were 8 ± 4 h and 15 ± 2 h, respectively. Hence, the FucciFT2 system was chosen as the best, and could be used to visualize transitions between the G1 and S/G2/M phases of the cell cycle.

### 2.8. Characterization of the mTagFT Timer Crystal Structure

To understand the effect of mutations introduced during mTagFT’s development on its blue-to-red timer properties, we determined the crystal structure of the mTagFT red form with 2.9 Å resolution (Figure 8 and Appendix A). mTagFT has a typical β-barrel overall structure with the chromophore formed by ^65^LYG^67^ amino acids located in the central α-helix (Figure 8a). There are four protein subunits per asymmetric unit with very similar folds (RMSD does not exceed 0.4 Å^2^), which do not form any stable oligomeric states according to crystal contact analysis.

Despite the crystal twinning and relatively low resolution of the structure, it was possible to identify chromophore molecules in all four subunits of the asymmetric unit. There was a lack of electron density around the tyrosine moiety of the chromophores, probably reflecting their flexibility and/or poor crystal quality (Figure 8c). Unlike the Fast-FT blue-to-red fluorescent timer [27], mTagFT has an intact chromophore without clear signs of covalent bond cleavage and contains serine residues in the 148 and 168 positions (enumeration follows to Figure 1) near the phenolic hydroxyl of the chromophore (Figure 8b). Usually, in RFPs, the formation of an H-bond between the chromophore phenolic hydroxyl and S148 hydroxyl group stabilizes a cis-chromophore configuration. In contrast, the H-bond with the S165 hydroxyl group is favorable for a trans-chromophore. Hence, in the case of mTagFT, both types of chromophores might be formed. Although the poor electron density around the tyrosine moieties of the chromophores does not allow us to unequivocally determine the cis or trans configuration based only on the structural data (Figure 8c), we can judge the configuration of the chromophore according to biochemical data. First, the excitation and emission spectra of the mTagFT red form are blueshifted compared to those of the mRubyFT red form, which has a cis-chromophore (Table 1) [13]. Similarly, excitation and emission spectra of the mKate RFP, with a cis-chromophore [28] are redshifted compared to the spectra of the parental TagRFP RFP, with a trans-chromophore [29]. In addition, the red form of the mTagFT/S165A mutant has a 26 nm redshifted emission maximum compared to mTagFT (see Section 2.6), indicating that the S165A mutation is likely to cause the transformation of the chromophore from the trans to the cis configuration. Hence, biochemical data support the formation of the trans-chromophore in the mTagFT protein.

Then, we analyzed the contacts of the mTagFT chromophore with the immediate environment (Figure 8b). All four chromophore molecules have similar coordinations in the protein, involving four direct hydrogen bonds to R69, W94, R96, and E222. Because the hydroxyl group of the chromophore is likely in the trans-configuration, it is additionally could be stabilized by H-bonds to the hydroxyl groups of S165 and/or S148. In addition, the phenolic group of the chromophore in the trans-configuration could be stacked with the positively charged H203, which can stabilize the negative charge of the phenolic hydroxyl group.

To elucidate why mTagFT has blue and red forms, we compared chromophores and their environments for mTagFT and its predecessor, TagRFP RFP (PDB ID–3M22). Compared to TagRFP, mTagFT has six mutations that are inner to the β-barrel: M14I, I46V, V65L, N148S, Q220L, and A224S (Figure 9a, in blue, and Figure 1). Many of them are on the side of the acylimine group.

The last four mutations are the same as in mRubyFT, and therefore, they are probably necessary for the timer phenotype. L65 is the first amino acid of the chromophore that forms a triplet. It was shown earlier that bright blue fluorescent proteins such as mTagBFP [30] and mTagBFP2 [26], obtained from TagRFP RFP, contained leucine in the first position of the chromophore. In addition, the best blue variants of the RFPs mCherry, HcRed1, M355NA, and mKeima, named mCherry-Blue, HcRed1-Blue, M355NA-Blue, and mKeima-Blue, respectively, have leucine or histidine in the first position of the chromophore tripeptide [30]. The mRubyFT timer with a bright blue form also contained the LYG chromophore triplet [13]. Hence, leucine in the first position of the chromophore (M65L mutation) in mTagFT is responsible for the formation of its bright blue form. The Q220L mutation is favorable for stabilization of the blue form, since the hydrophobic side chain of L220 lies in the same hydrophobic pocket as the side chain of L65 (Figure 9a). This pocket is additionally formed by side chains of M44, F58, F64, A61, and L205, and seems to be strengthened by the replacements at I14 and V46. The N148S mutation in mTagFT had an unclear effect on the chromophore configuration. Although the side chain of S148 should interact with the phenolic hydroxyl group of the chromophore and stabilize its cis-configuration via an H-bond (Figure 8b), we did not observe the cis-configuration of the chromophore either in the crystal structure or in biochemical experiments, which argues for the trans-configuration. The A224S mutation in mTagFT seems not to lead to the formation of any H-bonds involving this side chain. At the same time, in the vicinity of S224, the H-bond between H203 and E222, present in TagRFP, is lost in mTagFT. This is caused by the formation of an H-bond between the side chain of E222 and the imidazolinone nitrogen of the chromophore (Figure 8b). Notably, mutagenesis of S222 in mRubyFT (S224 in mTagFT) showed that this amino acid is important for timer characteristics and maturation [13]. Therefore, mTagFT contained three mutations, M65L, Q220L, and A224S, that might be associated with its blue-to-red timer properties. The additional two mutations, M14I and I46V, which are internal to the β-barrel, may stabilize the blue form of mTagFT.

Contact analysis demonstrated that mTagFT seems to be a monomer with weak dimeric contacts in the crystal. Compared to TagRFP, mTagFT has four mutations that are external to the β-barrel: N11I, M151K, Y153K, and C229P (Figure 1 and Figure 9a). Subunits in tetrameric fluorescent proteins, such as mKate (a derivative of TagRFP), are known to form two types of interfaces, IF1 and IF2, where IF1 is substantially weaker than IF2. Comparison to the mKate tetrameric structure (PDB ID–3RWA) demonstrated the complete loss of the IF2 interface in the case of mTagFT (Figure 9b). Three residues in mTagFT, substituted at positions 151, 153, and 229, are located directly on the IF2 interface. Taking into account an absence of the IF2 interface in mTagFT, this indicates the effective destruction of this interface due to the M151K, Y153K, and C229P mutations. This is in agreement with the fact that the M151K, Y153K, and C229S mutations introduced into PATagRFP allowed breaking of the IF2 interface and monomerization of the protein [14]. Notably, the crystal structure of mTagFT does not have a canonical IF1 interface (Figure 9b); instead, there is another weak interface (approximately 7 HB/SB), with one disulfide bond made by C118, of two adjacent subunits, where the N11I mutation is located. Notably, the C118S mutation did not alter the slight admixture of oligomers found in addition to the monomer fraction (see Section 2.1), indicating that this interface does not exist in solution. The hydrophobic side chain of I11 is oriented toward the protein surface and could alter possible hydrogen bonds with the neighboring subunit, which may also weaken this interface. Hence, all external mutations in mTagFT compared to TagRFP contribute to its monomerization.

In summary, we related mutations found in the mTagFT timer to its X-ray structure.

### 2.9. Directed Mutagenesis of the mTagFT Timer at Key Positions

We further performed directed mutagenesis of the mTagFT timer at the amino acid residue positions 16, 44, 65, 148, 165, 203, 220, and 224 (enumeration follows Figure 1), which are around the chromophore according to the X-ray structure, and characterized the spectral properties of the mutants (Table 3 and Appendix A).

Mutations at positions 16 and 44 resulted in a far-redshift in mPlum [31] and Neptune RFPs [25], so we anticipated that L16E and M44C or M44Q mutations could shift the excitation/emission maxima of the mTagFT timer to far-red wavelengths. The mTagFT L16E mutant was nonfluorescent. The M44C mutation did not notably affect the spectral properties of either the blue or red form (Table 3 and Appendix A). However, the M44Q mutation shifted the absorption/excitation/emission maxima of the blue and red forms to far-red wavelengths by 6/9/7 and 10/1/10 nm, respectively (Table 3 and Appendix A). The shifts in excitation and emission maxima for the mTagFT/M44Q mutant were similar to those previously observed for the Neptune and mPlum RFPs [25,31]. Hence, contacts of the residue at position 44 with the acylimine group of the mTagFT chromophore were important for the bathochromic shift of its excitation/emission spectra.

Positions 65 and 220 were suggested to be the key for the timer-like phenotype in the case of the mRubyFT timer; however, mutagenesis of the mRuby2 protein did not confirm this suggestion [13]. Compared to the mRubyFT timer, we found the same substitutions for the TagFT and mTagFT proteins (Figure 1), so we restored these mutations in the mTagFT timer. The mTagFT/L220Q mutant and mTagFT/L65M/L220Q double mutant were not fluorescent (Table 3). The mTagFT/L65M mutant preserved the blue-to-red transition over time in bacterial cells at 37 °C, and its blue and red forms had spectral properties similar to those of the original mTagFT protein (Table 3 and Appendix A). In the absorption spectrum of the mTagFT/L65M mutant, we observed a new absorption peak at 507 nm with excitation/emission maxima at 590/522 nm, which were attributed to the GFP-like green form (Table 3 and Appendix A). Therefore, position 65 was not critical for the appearance of the blue-to-red timer-like phenotype of the mTagFT protein, but was important for the efficient formation of the blue form.

The replacement of the residue at position 148 with the bulky Ile or Phe in the mRubyFT timer stabilized the blue form and blocked the formation of the red form [13]. The introduction of the S148I mutation into the mTagFT protein also led to the formation of a blue fluorescent protein with spectral properties similar to those of the blue form of the mTagFT protein (Table 3 and Appendix A); the freshly purified protein did not exhibit red fluorescence. Hence, position 148 was important for stabilizing the blue form of the mTagFT protein.

Position 165 was important for the formation of the red form of the mRubyFT timer [13]. According to the X-ray structure of the mTagFT timer, Ser165 is supposed to form a hydrogen bond with the hydroxyl group of the chromophore and stabilize the red chromophore in a trans-like configuration (Figure 8). The S165A mutation resulted in a notable bathochromic shift of the absorption/excitation/emission maxima of the red form of mTagFT by 9/18/26 nm, respectively (Table 3 and Appendix A). These data assume that the S165A mutation might destabilize the trans-like configuration of the red chromophore and result in the formation of the cis-like configuration of the red chromophore, supporting the hypothesis that the chromophore of the mTagFT timer is in the trans-like configuration.

The S165A mutation did not affect the spectral properties of the blue form (Table 3 and Appendix A). The M44Q mutation in contact with the acylimine group of the chromophore affected the spectral properties of both the blue and red forms (see above). Hence, it can be concluded that the residue at position 165 is responsible for adjusting the spectral properties of the red form of the mTagFT timer rather than the blue form, confirming that the phenolic ring of the blue chromophore is not conjugated to the imidazolinone ring.

According to the mTagFT structure, the His residue at position 203 was in stacking interactions with the chromophore (Figure 8). The H203I mutation slightly affected the spectral properties of the red form of the mTagFT timer and resulted in the disappearance of the fluorescence of the blue form (Table 3 and Appendix A). The replacement of the His203 residue with an aromatic Tyr residue in the mRubyFT protein preserved the blue fluorescence of the mRubyFT timer [13]. Hence, stacking interactions between the chromophore and the residue at position 203 ensured the fluorescence of the blue form of the mTagFT timer.

The S224A mutation in the mRubyFT and FastFT timers was not crucial for the formation of the blue and red forms [13,27]. The mTagFT/S224A mutant also revealed blue fluorescence, which transformed into red fluorescence over time; the blue and red excitation/emission maxima were shifted by -12/-1 and -5/11 nm, respectively (Table 3 and Appendix A). Characteristic maturation times for the blue and red forms of the mTagFT/S224A mutant were observed at 1.4 and 4.9 h, respectively. Hence, the S224A mutation accelerated the maturation of the blue and red forms by 3.4-fold and 1.6-fold, respectively. The analogous S217A mutation in the FastFT and S224A in mRubyFT timers also affected their characteristic maturation times [13,27]. Hence, the S224A mutation accelerated the maturation of the mTagFT timer and affected the spectral properties of its blue and red forms.

In summary, structural and mutagenesis data revealed the impact of the mutations on the properties of the mTagFT timer, and could help in the further development of the blue-to-red and blue-to-far-red fluorescent timers in future.

## 3. Materials and Methods

### 3.1. Cloning of Bacterial Plasmids, Mutagenesis and Library Screening

The cloning of bacterial plasmids, mutagenesis, and library screening were performed as described previously [13], using the primers listed in Appendix A. For details, please see the Appendix A.

### 3.2. Protein Purification and Characterization

Proteins were purified and characterized as described in [13]. For details, please see the Appendix A.

### 3.3. Obtaining Stable Cell Lines

Stable cell lines were obtained according to [32]. Briefly, the plasmid pSBbi-GN-FT-Hygromycin or pSBbi-GN-FT-Puromycin (derived from pSBbi-GN Plasmid Addgene (Watertown, MA, USA) #60517) was mixed (10:1) with the plasmid pCMV(CAT)T7-SB100 (Plasmid Addgene (Watertown, MA, USA) #34879) and transfected into HeLa cells. Twenty-four hours after transfection, the cells were subjected to hygromycin (300 µg/mL final) and/or puromycin (2 µg/mL) antibiotic selection for two to ten days. Cells were trypsinized and resuspended in DPBS buffer, filtered through a 70 µm filter, and sorted using a BD FACSAria Fusion flow cytometer (Becton, Dickinson and Company BD Biosciences, San Jose, CA, USA). Approximately 150,000 green/far-red fluorescent HeLa cells stably expressing mNeptusFT2-hGeminin were sorted in one well of a 6-well plate using a 10% gate. The 96 blue/red and nongreen/non-far-red fluorescent single cells expressing FT-hCdt_1-100_ and mNeptusFT2-hGeminin were individually sorted in 96-well plates, and three clones with the fastest growth rate and brightest fluorescence were selected for each timer for further characterization.

### 3.4. Isolation of Adeno-Associated Viral Particles

For the preparative production and isolation of adeno-associated viral particles (rAAVs), we plated approximately two million HEK293T cells on six 15-cm diameter Petri dishes in DMEM (Paneco, Moscow, Russia) containing 10% FBS (Paneco, Moscow, Russia), GlutaMax I and streptavidin/penicillin (Paneco, Moscow, Russia) in a CO_2_ incubator (5% CO_2_). The cell monolayer was then expanded to 80% confluence for approximately 1 to 2 days. Cells were transfected with a mixture of the following plasmids: pAAV-DJ (6 × 28 µg), pAAV-Helper (6 × 28 µg), and pAAV-arc1-TagFT-3xNLS (6 × 28 µg).

A total of 11 mL of solution A, containing 6 × 3 × 28 μg of DNA mixture and 2733 μL of 1 M CaCl_2_; and 11 mL of solution B containing 2× HBSS in water, were prepared. Next, approximately 1/8 of solution A was added to solution B while gently stirring, followed by incubation of the mixture at room temperature for 20 min. Five to 10 min before the end of incubation, the cells were changed to 23 mL of DMEM containing streptomycin-penicillin (100 μg/mL) without serum. The incubated mixture was then added to the cells and mixed gently with the medium. The culture was incubated in a CO_2_ incubator (37 °C, 5% CO_2_) for 16 h. After incubation, the medium was replaced with 23 mL of DMEM containing 10% BSA, GlutaMax-I, and streptomycin-penicillin (100 μg/mL).

After 2 days, the cells were separated from the surface by incubation with 2 mL of trypsin (0.25%) for 5 min at 37 °C, followed by resuspension in 2 mL of DPBS and centrifugation at 200× *g* for 2 min. Cells from one 15-cm dish were further resuspended in 10 mL of 100 mM NaCl, 20 mM Tris-HCl, pH 8.0 buffer containing 0.5% sodium deoxycholate, and benzonase (final concentration of 25 units/mL). The cells were then lysed for 1 h at 37 °C and 220× *g*. Cell residues were then removed by centrifugation at 3000× *g* for 15 min, and the supernatant was transferred to 50 mL tubes with 0.5 mL of heparin resin. Viral particles were bound to the resin for 1 h under stirring at room temperature. The resin with bound particles was precipitated by centrifugation at 3500× *g* for 2 min, and the supernatant was removed. The resin was then washed with 6 mL of 100 mM NaCl, 20 mM Tris-HCl, and pH 8.0 buffer, followed by centrifugation at 3500× *g* for 1 min and aspiration of the supernatant. The resin was washed again with 6 mL of 100 mM NaCl, 20 mM Tris-HCl, and pH 8.0 buffer, followed by centrifugation at 3500× *g* for 1 min and aspiration of the supernatant. The resin was then transferred to the column and washed with 1 mL of 100 mM NaCl, 20 mM Tris-HCl, and pH 8.00 buffer. rAAV particles were eluted in 2.5 mL of 500 mM NaCl, 20 mM Tris-HCl, and pH 8.00 buffer. Viral particles from the solution were precipitated by ultracentrifugation for 24 h at 80,000× *g* and 15 °C in 1.5 mL tubes containing 0.6 mL of particle eluate. The viral particle precipitate was resuspended in 20 to 25 μL of 500 mM NaCl, 20 mM Tris-HCl, and pH 8.00 buffer.

Next, the viral particle titer was determined by transducing HEK293T cell cultures plated on glass-bottomed dishes (Matteck, Ashland, MA, USA) with different amounts of viral particles. For primary neuronal culture experiments, viral particles with a titer of at least 2 × 10^8^ virulent particles/mL were used.

### 3.5. Extraction, Cultivation and Transduction of Dissociated Neuronal Cultures

Dissociated hippocampal neuronal cultures obtained from nine newborn (P0) pups of the B6.Cg-Tg(Fos/EGFP)1-3Brth/J mouse line (in which the EGFP gene is under the c-fos promoter) were planted on 33 mm Matteck glass-bottomed dishes (Matteck, Ashland, MA, USA) or coverslips and maintained as follows. Matteck glass-bottom cups or coverslips were first sterilized by washing in 70% ethanol, followed by UV irradiation in a closed laminar for 1 h, and then treated with 0.05% polyethyleneimine to ensure neuronal adhesion to the surface. Next, the newborn mouse was decapitated using scissors. Under sterile conditions (under laminar flow), the brain was extracted and immediately placed in a cooled dissection buffer. The hippocampus or cortex was dissected under the microscope using tweezers and a scalpel from two hemispheres in series, freed from the vascular plexus, and placed in a new cup with cold-cooled dissection buffer. After the hippocampi/cortexes were isolated from all mice, they were transferred to a new cup with forceps and shredded with a scalpel in a drop of buffer. The shredded tissue was transferred to a test tube with 2 mL of 0.25% trypsin (Gibco, Oxford, UK) prewarmed to 37 °C, and placed in a 37 °C thermostat for 20 min. Full culture medium (Neurobasal medium Neurobasal (Gibco, Oxford, UK), combined with bioactive supplement B27 (Gibco, Oxford, UK) and glutamine (Gibco, Oxford, UK)), was prewarmed to 37 °C and added to the tube with trypsinized tissue. The mixture was carefully resuspended, and the cells were then precipitated using a centrifuge at 1800× *g* for 5 min. The supernatant was removed, and the cells were resuspended in 9 mL of complete culture medium and then precipitated using a centrifuge at 1800× *g* for 5 min. The supernatant was removed, and the cells were resuspended in 1 mL of complete culture medium. The cells were then counted, and the cell concentration was adjusted to three million/mL. Cells in a volume of 50 μL were plated onto glass-bottomed or glass dishes pretreated with polyethyleneimine or polylysine. The cell culture density was at least 1200 cells/mm. The glass-bottomed or glass dishes were placed for 30 min in a 37 °C CO_2_ incubator. After incubation, 1 mL of prewarmed 37 °C medium was added dropwise to the glass-bottomed or glass dishes, and the cells were placed in the CO_2_ incubator. Culture viability was maintained under CO_2_ incubator conditions at 37 °C and a gas mixture of 95% air and 5% CO_2_. The culture liquid was replaced with new medium (half of the volume was replaced with an equal volume of new medium) one day after plating cells on the dishes, and then once every 3 days.

Transduction of neuronal cultures was performed on day 4 after plating the cultures by injecting 1 μL of a solution of viral particles carrying the TagFT timer gene.

### 3.6. Purification of the mTagFT Protein for Crystallization

Preparative mTagFT protein purification was performed as described in [13]. For details, please see the Appendix A.

### 3.7. Protein Crystallization

An initial crystallization screening of mTagFT was performed with a robotic crystallization system (Rigaku, Woodlands, TX, USA) and commercially available 96-well crystallization screens (Hampton Research, Aliso Viejo, CA, USA and Anatrace, Maumee, OH, USA) at 15 °C using the sitting drop vapor diffusion method. The protein concentration was 8 mg/mL in the following buffer: 50 mM sodium phosphate, 40 mM NaCl pH 7.5. The initial conditions were optimized by the hanging-drop vapor-diffusion method in 24-well VDX plates. The crystals were obtained within 6 months under the following conditions: 0.2 M lithium sulfate, 0.1 M Bis-tris pH 6.5, and PEG 3350 27%.

### 3.8. Data Collection, Processing, Structure Solution, and Refinement

The mTagFT crystals were treated and tested, and data were collected and analyzed, as described previously [13]. For details, please see the Appendix A.

### 3.9. Structure Analysis and Validation

Visual inspection, comparison, and superposition of the structures were performed as described previously. For details, please see the Appendix A.

### 3.10. Mammalian Plasmid Construction

Mammalian plasmid construction was performed as described in [13]. For details, please see the Appendix A.

### 3.11. Mammalian Live Cell Imaging

Confocal imaging of live cells was performed as described in [13]. For details, please see the Appendix A.

### 3.12. Statistics

To estimate the significance of the difference between two values, we used the Mann–Whitney rank sum test and provided *p* values (in brackets throughout the text) calculated for the two-tailed hypothesis. We considered the difference significant if the *p* value was <0.05.

## 4. Conclusions

In conclusion, we developed and characterized a set of novel true blue-to-red fluorescent timers, TagFT and mTagFT, derived from TagRFP; and novel tandem blue-to-red mTsFT and green-to-far-red mNeptusFT2 fluorescent timers, based on mTagBFP2 BFP and mScarlet RFP fusion, and sfGFP GFP and mNeptune far-red FP fusion, respectively. Using these timers, we demonstrated different biological applications. A split version of the TagFT timer allowed the detection of interactions between proteins in mammalian cells. The TagFT timer under the control of the minimal arc promoter successfully visualized the induction of immediate-early genes in neuronal cultures. Based on the combination of true and tandem timers, we developed the FucciFT2 system to accurately visualize transitions between phases of the cell cycle.

For the first time, we compared true and tandem fluorescent timers under the same experimental conditions. Compared to the true blue-to-red timers mRubyFT, mTagFT, TagFT, MediumFT, and FastFT, the blue-to-red tandem timer mTsFT was advantageous in terms of the increased (by more than 24-fold) brightness of the red form both in vitro (Table 1) and in mammalian cells (Figure 3b,c and Figure 7b,c). In contrast to true blue-to-red timers, the tandem mTsFT timers did not suffer from blue-to-red photoconversion induced by illumination with blue or cyan light (Figure 3 and Appendix A). However, in contrast to the true TagFT timer, the tandem mTsFT timer is not appropriate for the generation of a split version and targeting of the engram neurons. Additionally, in contrast to true blue-to-red timers, fusions of the tandem mTsFT and mNeptusFT2 timers with the hCdt1-100 protein were toxic to cells or did not degrade in the S/G2/M phase of the cell cycle, respectively (Section 2.7). The toxicity and poor degradability of tandem timers in mammalian cells is probably connected with the large size of the tandem molecule. The large size of the tandem timers is also not favorable for packaging into rAAV particles.

TagFT, and especially mTagFT, were susceptible to blue light-induced blue-to-red photoconversion (Figure 3d–f), similar to the mRubyFT and FastFT timers [13]. In addition to photoconversion, the red forms of the TagFT and mTagFT timers were also prone to photochromism (Figure 2f and Appendix A). These are highly undesirable properties, which can seriously interfere with the quantification of timing red/blue ratios. Hence, imaging of the blue and red forms should be performed at the minimal possible excitation light power, and the red form should be imaged before imaging of the blue form.

The generated split version of the TagFT timer has a great potential for visualization of protein—protein interactions, since it matures at 37 °C, and it potentially provides information about the time of the beginning of this interaction. However, the split TagFT should be characterized better in the future, in terms of its efficiency of complementation and comparative brightness to the corresponding non-split version. We also suggest that the dimer formation observed for the TagFT protein (Figure 2e) should decrease the fluorescence contrast between interacting and non-interacting proteins. However, split version of the monomerized variant of the TagFT protein, mTagFT, did not mature at 37 °C in mammalian cells. Finally, to show that such split systems can indeed provide quantitative information on the timing of a protein-protein interaction would require a controllable protein-protein interaction system as proof of concept.

The commonly used methods for labeling the two populations of neurons involved in different episodes of cognitive activity or engrams were mainly feasible ex vivo, using some histological methods of sectioning the inactive deceased fixed brain [33]. The method of labeling neuronal engrams based on the TagFT timer under the control of the minimal arc promoter is appropriate for delivery into the mouse brain using rAAVs, could be used for engram marking in the living brain in vivo, and does not require staining with antibodies. 

In previous studies, a number of residues were suggested for the development of a bright blue fluorescent probe from red fluorescent proteins [13,30]. The mutations found in permanently blue fluorescent mTagFT/S148I mutants support this rule.

The FucciFT2 system developed in this study is the first example of the utilization of fluorescent timers for visualization of the cell cycle. In comparison to the regular Fucci system, FucciFT2 makes it possible to determine the duration of the cell in the G1 and S/G2/M phases in a single image (Figure 7e,f), and there is no need for prolonged continuous imaging. In this regard, the FucciFT2 system is appropriate for cell cycle analysis using flow cytometry.

We assume that the blue forms of the TagFT and mTagFT timers have a TagBFP-like chromophore structure [29] since they share the same LYG-chromophore tripeptide and have similar spectral characteristics (Table 1). Structure-guided mutations introduced in the mTagFT protein (Section 2.5 and Section 2.6) revealed that it is possible to adjust its spectral properties for the future development of a blue-to-far-red fluorescent timer.

## Figures and Tables

**Figure 1 ijms-24-03279-f001:**
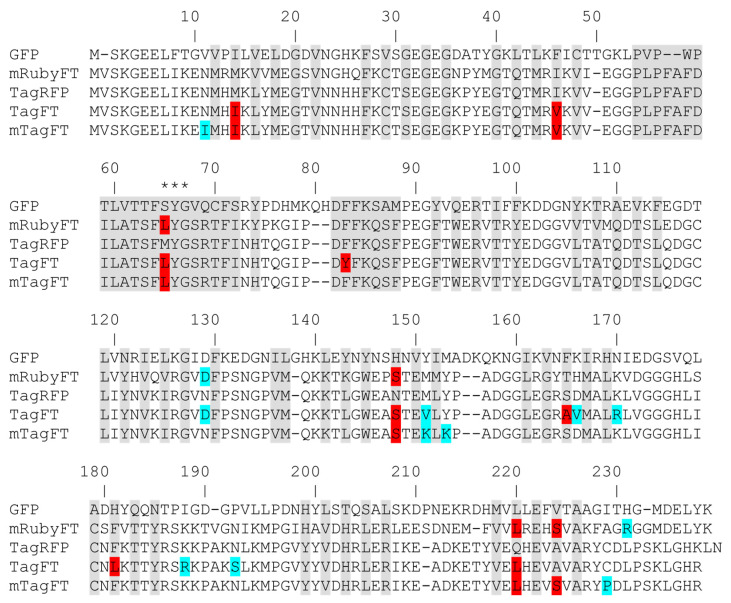
Amino acid sequence alignment of the timers TagFT, mTagFT, and mRubyFT and the fluorescent proteins GFP and TagRFP. Alignment numbering follows that of *Aequorea victoria* GFP. The residues inside the β-barrel are in gray. Asterisks (***) indicate the chromophore-forming tripeptide. Internal and external mutations in the TagFT, mTagFT, and mRubyFT timers, relative to the TagRFP and mRuby2 progenitors, are highlighted in red and cyan, respectively.

**Figure 2 ijms-24-03279-f002:**
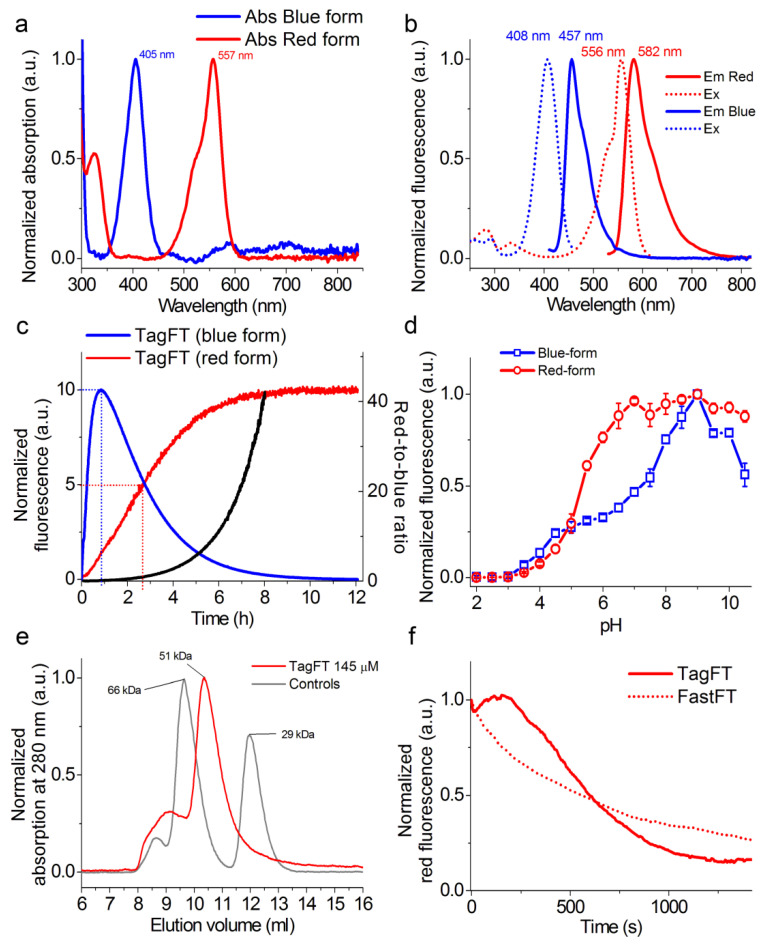
In vitro properties of the purified TagFT protein. (**a**) Absorption spectra of the blue and red forms of the TagFT protein in PBS buffer at pH 7.40. (**b**) Excitation and emission spectra of the blue and red forms of TagFT in PBS buffer at pH 7.40. (**c**) Maturation of blue and red forms of TagFT in PBS buffer at pH 7.40 and 37 °C. The red-to-blue ratio (black curve) was calculated according to the red and blue fluorescence time dependences normalized to 100. (**d**) Fluorescence intensity of the blue and red forms of TagFT as a function of pH. Three replicates were averaged for analysis. Error bars represent the standard deviation. (**e**) Fast protein liquid chromatography of the TagFT protein. TagFT eluted in 20 mM Tris-HCl (pH 7.80) and 200 mM NaCl buffer. The molecular weight of TagFT was calculated from a linear regression of the dependence of logarithm of control molecular weights vs. elution volume. (**f**) Photostability of red forms of TagFT and control FastFT timers under continuous wide-field imaging using a mercury lamp (9 mW/cm^2^ 550/25 nm light power before objective lens).

**Figure 3 ijms-24-03279-f003:**
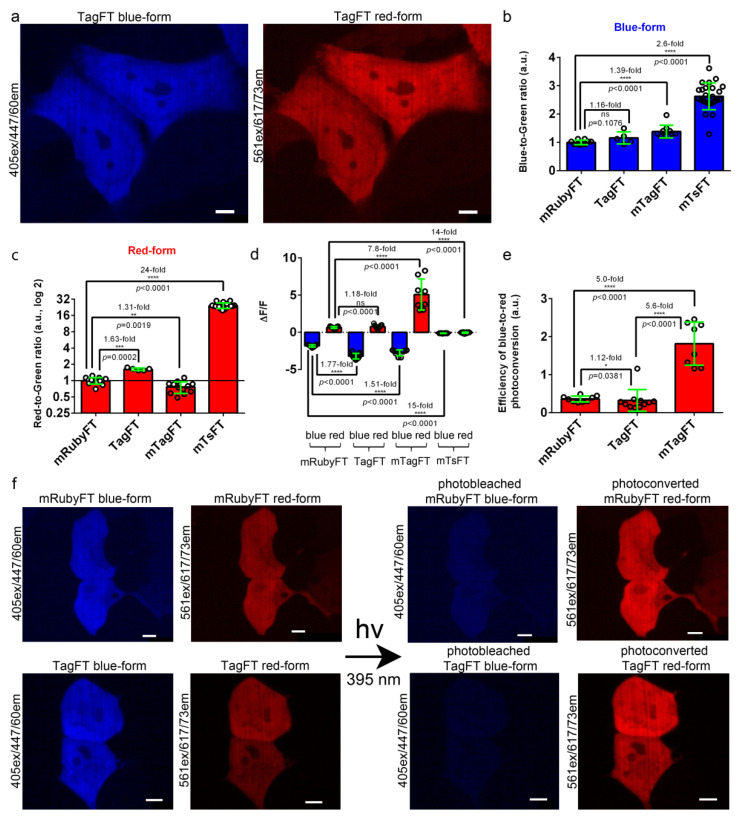
Comparison of the brightnesses and blue-to-red photoconversions of the TagFT, mTagFT, mTsFT, and control mRubyFT timers in live mammalian cells. (**a**) Confocal images of live HeLa cells expressing the TagFT-P2A-EGFP fusion. P2A is a self-cleavable peptide. Blue (405 ex and 447/60 em) and red (561 ex and 617/73 em) fluorescence channels are shown for expression lasting for 72 h. (**b**,**c**) The averaged brightness of the blue (**b**, 24 h after transfection) and red forms (**c**, 72 h after transfection) of the TagFT, mTagFT, mTsFT, and control mRubyFT timers in HeLa cells normalized to the brightness of the EGFP expressed in the same cell. Error bars are standard deviations across 5–24 cells. (**d**) The mean ΔF/F values for the photobleached blue forms and photoconverted red forms of TagFT, mTagFT, mTsFT, and control mRubyFT timers expressed in live HEK293T cells 24 h after transfection. The pulse of 395/25 nm light (0.338 mW/cm^2^ power measured before the 60× oil objective lens) lasted for 1 min. Error bars are standard deviations across 8–11 cells. (**e**) The efficiency of the blue-to-red photoconversion with 395/25 nm light for 1 min was calculated as ΔF/F_red_/ΔF/F_blue_. (**f**) Confocal images of live HEK293T cells expressing mRubyFT-P2A-EGFP or the TagFT-P2A-EGFP fusion. Blue (405 ex and 447/60 em) and red (561 ex and 617/73 em) fluorescence channels before and after continuous irradiation with 395/25 nm light for 1 min are shown. Protein expression lasted 24 h. The contrast settings were the same for red and blue images. Images were acquired 72 h (**a**) or 24 h (**f**) after transfection. (**b**–**e**) The *p* values show significant differences between the respective values. ****, *p* value is <0.0001. **, *p* value is <0.01. ***, *p* value is <0.001. *, *p* value is <0.05. (**a**,**f**) Scale bars: 50 µm.

**Figure 4 ijms-24-03279-f004:**
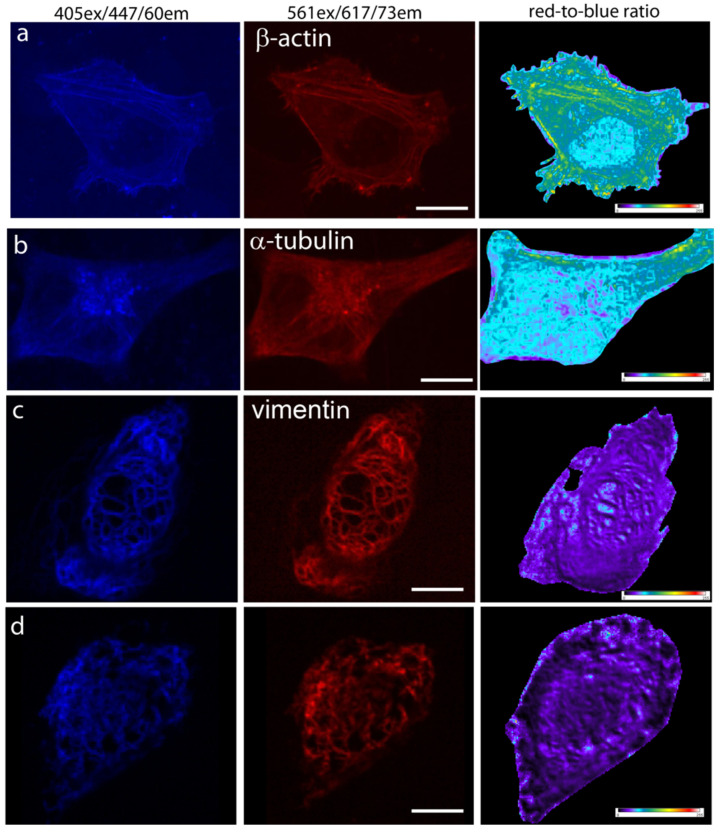
Confocal imaging of the mTagFT timer in fusions with cytoskeleton proteins in live mammalian cells. Confocal images of HeLa Kyoto cells in blue (405 ex and 447/60 em) and red (561ex and 617/73em) channels and red-to-blue ratio 24 or 72 h after transfection with (**a**) pmTagFT-β-actin, (**b**) pmTagFT-α-tubulin, (**c**) pVimentin-mTagFT, or (**d**) pVimentin-TagFT plasmids. Scale bar: 10 µm. For blue-to-red ratio image calculation, the background was subtracted, and a ratio image was generated using ImageJ software; the background ratio was manually cut around the cells.

**Figure 5 ijms-24-03279-f005:**
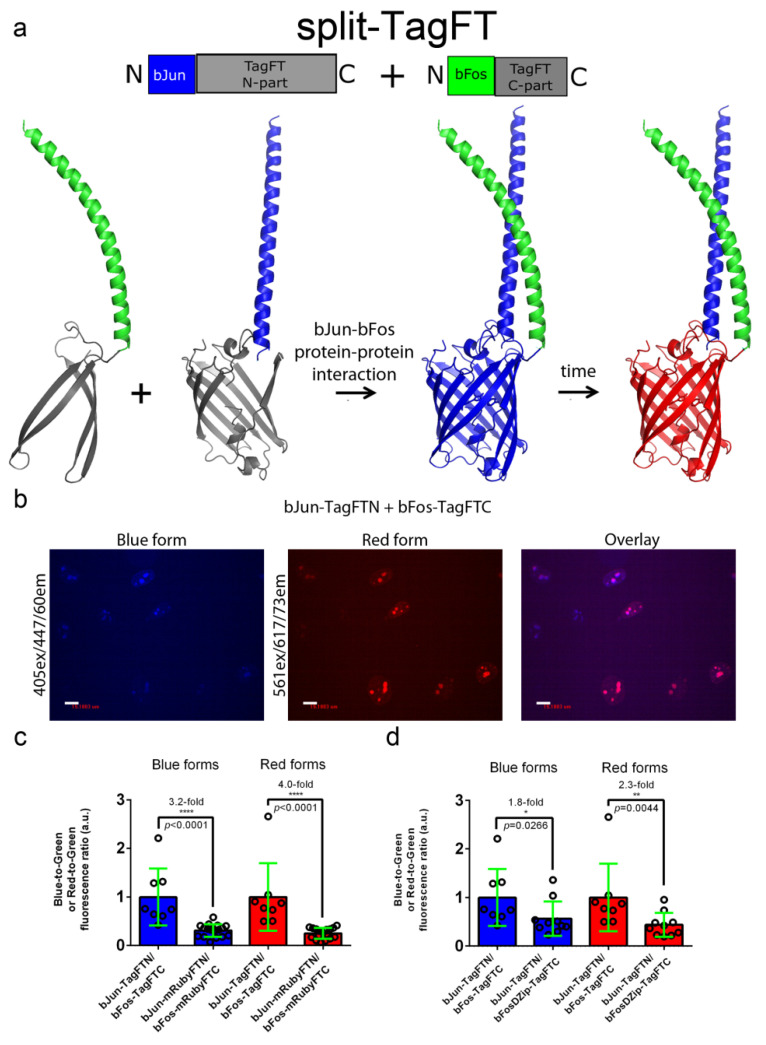
Characterization of the split TagFT version in mammalian cells. (**a**) Scheme of split mTagFT in fusion with bJun (in blue) and bFos (in green) heterodimerizing proteins, in which heterodimerization facilitates TagFT trimer assembly with formation of the blue form followed by conversion to the red form over time. (**b**) Confocal images of HeLa Kyoto cells in blue (405ex and 447/60em) and red (561ex and 617/73em) channels and channel overlay 24 h after cotransfection with pAAV-CAG-bJun-TagFTN and pAAV-CAG-bFos-TagFTN plasmids. Scale bar: 19 µm. (**c**) Comparison of the brightnesses of bJun-TagFTN/bFos-TagFTC and bJun-mRubyFTN/bFos-mRubyFTC constructs expressed in live HeLa cells. (**d**) Comparison of the brightnesses of bJun-TagFTN/bFos-TagFTC and bJun-TagFTN/bFosΔZip-TagFTC constructs expressed in live HeLa cells. (**c**,**d**) Brightnesses of the blue and red forms normalized to the brightness of EGFP coexpressed in the same cells. ****, *p* value is <0.0001. **, *p* value is <0.01. *, *p* value is <0.05.

**Figure 6 ijms-24-03279-f006:**
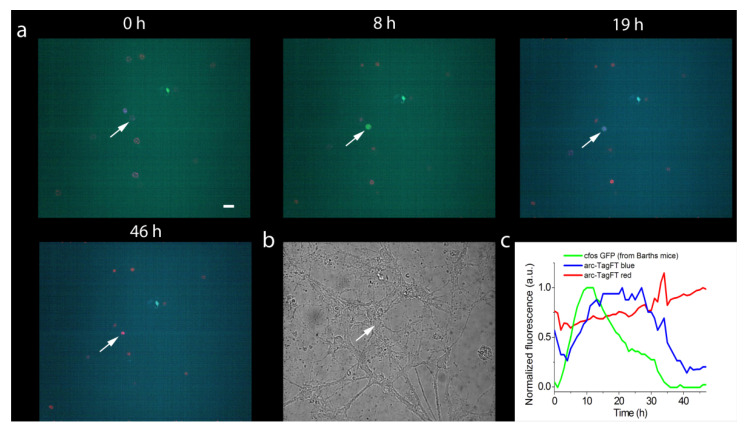
Chemical induction of the expression of the blue-to-red timer TagFT-3xNLS and the green protein EGFP from the promoters of the early genes arc1 and c-fos, respectively, in hippocampal culture. (**a**) Confocal images of cells in the superimposed blue–green–red fluorescence channel are shown at the indicated times after chemical stimulation of neuronal culture with 100 mM KCl (3 times 2 min each, 5 min apart). (**b**) Image of neuronal culture in visible light. (**c**) Dynamics of blue, green, and red fluorescence changes in the neuronal cell nucleus indicated by the arrow in panels **a** and **b**. The hippocampal culture was isolated from Barth line mice, which have a GFP under the control of the promoter of the early c-fos gene. On the fourth day after plating, the hippocampal culture was infected with recombinant adeno-associated viral particles carrying the arc1 promoter and TagFT timer gene *Arc1-TagFT-3xNLS*. Scale bar: 30 µm.

**Figure 7 ijms-24-03279-f007:**
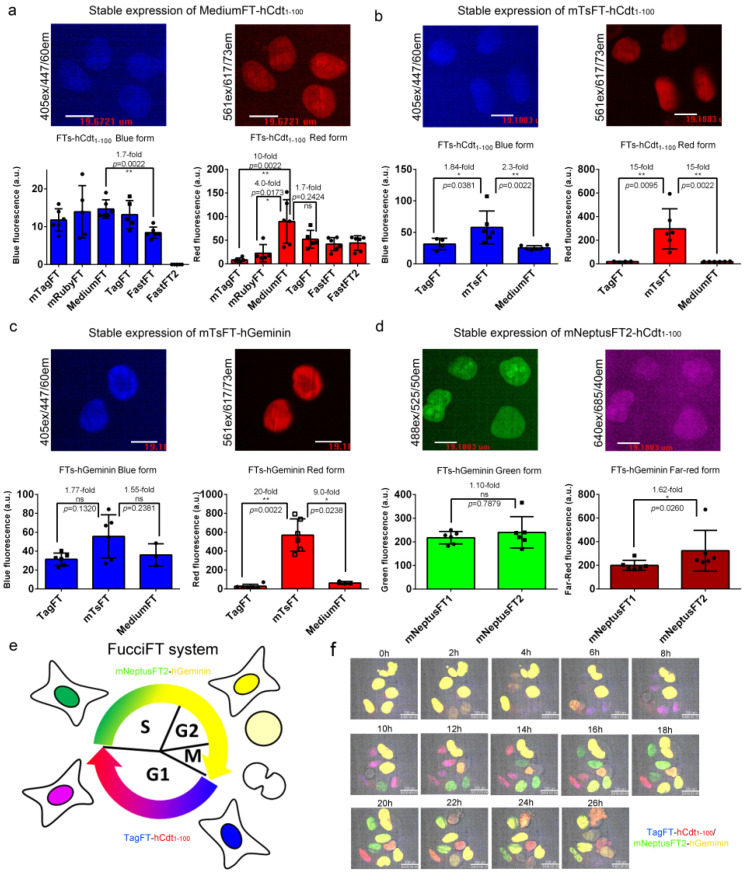
Brightness characterization of the timers in fusions with cycle-dependently degradable hCdt1-100 and hGeminin proteins in mammalian cells and visualization of the cell cycle using the FucciFT2 system, which includes blue-to-red TagFT-hCdt_1-100_ and green-to-far-red mNeptusFT2-hGeminin fusions. (**a**–**d**) Confocal images of HeLa cells stably expressing MediumFT-hCdt_1–100_ (**a**), mTsFT-hCdt_1–100_ (**b**), mTsFT-hGeminin (**c**), or mNeptusFT2-hCdt_1-–00_ (**d**). (**a**–**d**) Scale bar: 19 µm. The fluorescence intensities were quantified from images acquired using the same microscope settings. (**e**) Scheme describing the function of the FucciFT system, which colors the G1 and S/G2/M phases in blue/red and green/far-red (in yellow), respectively. (**f**) Confocal images of HeLa cells stably expressing the FucciFT2 system are shown in blue/green/red/far-red overlaid channels for blue-to-red TagFT-hCdt_1–100_ (in blue and red overlaid pseudocolors, respectively) and green-to-far-red mNeptusFT2-hGeminin fusions (in green and yellow overlaid pseudocolor, respectively). Scale bar: 30 µm. **, *p* value is <0.01. *, *p* value is <0.05. ns, *p* value is > 0.05.

**Figure 8 ijms-24-03279-f008:**
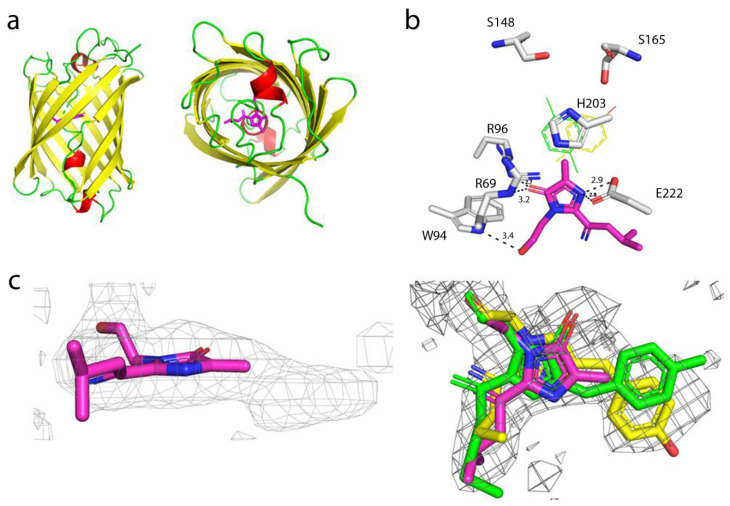
X-ray structure of the red form of the mTagFT protein. (**a**) Cartoon representation of the overall red mTagFT monomer. Chromophore, β-sheets, α-helixes, and loop regions are shown in pink, yellow, red, and green, respectively. The orientation of the panel on the right is rotated 90° around the horizontal axis with respect to that on the left. (**b**) The immediate environment of the red mTagFT chromophore. The imidazole ring of the chromophore is shown in magenta. Hydrogen bonds are depicted as dashed lines, and corresponding distances are labeled. Green and yellow tyrosine rings correspond to trans- and cis-configurations of TagRFP (PDB ID–3M22) and mRubyFT (PDB ID–7QGK), respectively. (**c**) The Polder electron density map (Fo-Fc) around the chromophore of the red mTagFT protein. The map is contoured at the 1.0 σ level and shown as gray mesh. The orientation of the chromophore on the right is rotated 90° around the horizontal axis with respect to that on the left. Similar to panel (**b**), green and yellow chromophores for trans- and cis-configurations are overlaid for clarity. Residue enumeration is shown in Figure 1.

**Figure 9 ijms-24-03279-f009:**
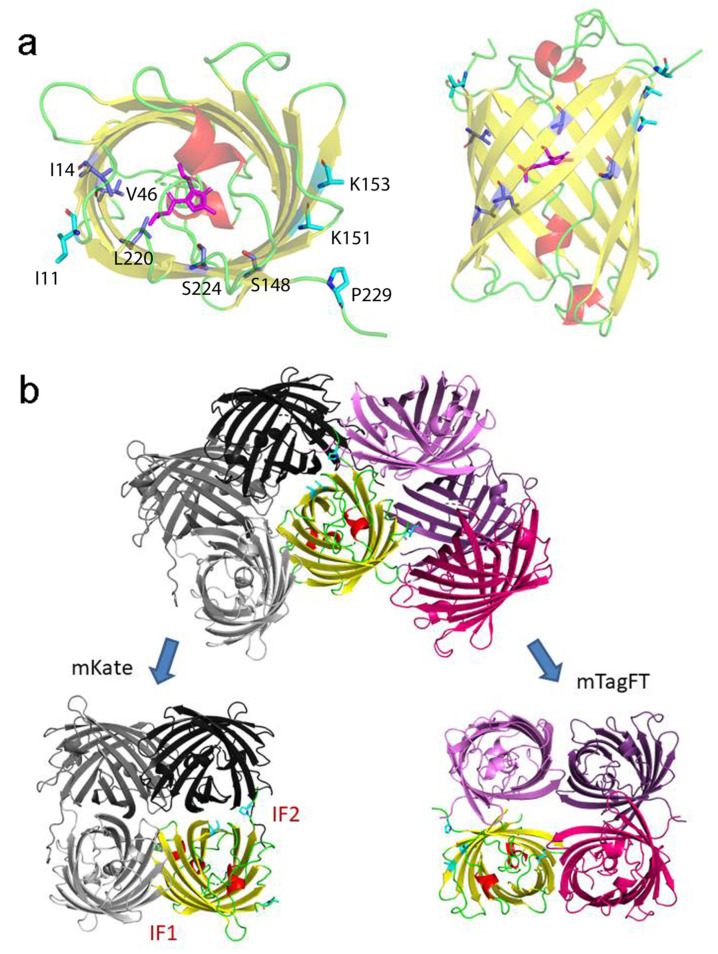
Location in mTagFT of internal and external mutations compared to TagRFP. (**a**) Cartoon representation of the overall red mTagFT monomer. Chromophore, β-sheets, α-helixes, and loop regions are shown in pink, yellow, red, and green colors, respectively. The orientation of the panel on the right is rotated 90° around the horizontal axis with respect to that on the left. Mutations in mTagFT compared to TagRFP internal and external to the β-barrel are shown in blue and light blue colors, respectively. (**b**) Overlay of mTagFT and mKate crystal structures. The IF1 and IF2 interfaces in the mKate tetramer are labeled. Tetramers are superposed by one protein chain shown in yellow only for mTagFT for clarity. Substituted residues are depicted for this subunit as in panel (**a**). Note that in the case of mTagFT, the tetramer is an artifact of crystal packing and is not stable but is shown for comparison with the mKate tetramer. Residue enumeration is shown in Figure 1.

**Table 1 ijms-24-03279-t001:** In vitro properties of the purified blue-to-red TagFT, mTagFT, and mRubyFT “true” timers and green-to-far-red mNeptusFT1, mNeptusFT2, and blue-to-red mTsFT tandem timers. ^a^—extinction coefficients for red forms were determined by the alkaline denaturation method or relative to the absorption at 280 nm (*), relative to the sfGFP absorption at 490 nm (**), or relative to the mScarlet absorption at 569 nm (***); extinction coefficients for blue forms were determined by the acid denaturation method or relative to the absorption at 280 nm (*). ^b^—quantum yields (QYs) for blue, red, and far-red forms were determined relative to mTagBFP2 (QY of 0.64), mCherry (QY of 0.22), and smURFP (QY of 0.18), respectively. ^c^—brightness was calculated as a product of quantum yield and extinction coefficient relative to the brightness of EGFP (QY of 0.6 and extinction coefficient of 56,000 M^−1^ cm^−1^). ^d^—characteristic times for the blue and red forms of TagFT, mTagFT, and mRubyFT correspond to the maximum of the blue fluorescence and half of the red fluorescence, respectively, at 37 °C; characteristic times for the green and far-red forms of mNeptusFT1 and mNeptusFT2 correspond to half of the green and far-red fluorescence, respectively, at 37 °C; characteristic times for the blue and red forms of mTsFT correspond to half of the blue and red fluorescence, respectively, at 37 °C. ^e^—data from reference [13]. ^f^—data from reference [15]. ^g^—data from reference [16].

Timer	Form	Abs, Ex/Em (nm)	ε (mM^−1^ cm^−1^) ^a^	QY ^b^	Brightness vs. EGFP ^c^(%)	Characteristic Times, h ^d^	p*K*_a_
mRubyFT ^e^	Blue	406, 408/457	96.8 (26.0) *	0.63	181 (49) *	5.7	3.9 ± 0.5
	Red	577, 582/624	60.0 (29.6) ^*^	0.086	15 (7.6) *	15	4.5 ± 0.1
TagFT	Blue	405, 408/457	33.4 (19.1) *	0.55	123 (31) *	0.83	7.3 ± 0.2
	Red	557, 556/582	281.0 (10.7) *	0.24	201 (7.7) *	2.7	5.32 ± 0.03
mTagFT	Blue	404, 406/461	69.5 (26.0) *	0.52	242 (40) *	4.7	4.8 ± 0.6
	Red	562, 557/590	66.0 (11.0) *	0.13	25 (4.2) *	8.0	4.91 ± 0.06
mNeptusFT1	Green	490, 490/514	83.3 ^f^	0.51	126	0.29	6.04 ± 0.08
	Far-Red	602, 602/662	20.4 **	0.22	13	7.5	5.7 ± 0.2
mNeptusFT2	Green	490, 490/514	83.3 ^f^	0.30	74	0.25	6.12 ± 0.06
	Far-Red	598, 598/650	84.5 **	0.27	68	4.4	5.66 ± 0.07
mTsFT	Blue	402, 402/456	85 ***	0.40	101	0.087	2.42 ± 0.05
	Red	569, 570/596	100 ^g^	0.70 ^g^	208	1.93	5.5 ± 0.2

**Table 2 ijms-24-03279-t002:** Comparison of the brightnesses of the blue-to-red TagFT, mTagFT, mTsFT, and control mRubyFT timers transiently expressed in HeLa mammalian cells. The blue and red fluorescences were normalized to the green fluorescence of EGFP expressed in the same cell in the FT-P2A-EGFP fusion construct. The brightnesses of the blue and red forms were registered 24 and 72 h after transfection, respectively.

Timer	Brightness vs. mRubyFT (%)	
	Blue Form	Red Form
TagFT	116 ± 22	163 ± 9
mTagFT	139 ± 23	77 ± 19
mTsFT	263 ± 48	2430 ± 280
mRubyFT	100 ± 8	100 ± 14

**Table 3 ijms-24-03279-t003:** Spectral properties of the purified mutants of the mTagFT timer. ^a^ Green form. ND, not determined. Enumeration corresponds to Figure 1.

Protein	Abs, Exc/Em, nm	
	Blue-Form	Red-Form
mTagFT	404, 406/461	562, 557/590
mTagFT/L16E	Nonfluorescent	Nonfluorescent
mTagFT/M44C	402, 405/465	561, 558/589
mTagFT/M44Q	410, 415/468	572, 558/600
mTagFT/L65M	404, 406/462 (507, 509/522) ^a^	ND, 556/584
mTagFT/S148I	402, 404/458	Nonfluorescent
mTagFT/S165A	ND, 405/461	571, 575/616
mTagFT/H203I	400, Nonfluorescent	561, 564/606
mTagFT/S224A	ND, 394/460	561, 552/601
mTagFT/L65M/L220Q	Nonfluorescent	Nonfluorescent

In comparison to TagFT, mTagFT also has two mutations, N129D and L231R (Figure 1), that are outer to the β-barrel.

## Data Availability

Data are contained within the article or Appendix A.

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
