# Peer review of "Blue-to-Red TagFT, mTagFT, mTsFT, and Green-to-FarRed mNeptusFT2 Proteins, Genetically Encoded True and Tandem Fluorescent Timers"

_ijms, 2023, doi:10.3390/ijms24043279_

Round 1

Reviewer 1 Report

The manuscript by Subach et al describes the generation of fluorescent timers from the TagRFP template. They describe two versions of the TagRFP fluorescent timer: TagFT and mTagFT, the latter of which was optimized for monomerization. The timers were characterized in terms of spectral properties, brightness and timing of blue to red conversion and an X-ray structure analysis is included for mTagFT. Next to TagFT timers also tandem timer constructs are presented with a fast maturing BFP or GFP and a slow maturing red or far red FP linked in one polypeptide, respectively. The overall performance of these tandem timers are compared to the recently described mRubyFT timer and the TagFT timers. Applications are shown for measuring protein-protein interactions with split-timer constructs, neuronal promotor activity and cell cycle progression.  

Main points:

1)    The organization of the paper is severely suboptimal.

The manuscript contains many different FT constructs and different applications that are not presented in a clear way. For instance, the authors recently have described the development of mRubyFT (ref 13). This was the best performing blue-to-red timer according to paragraph 4 of the introduction (section 1). So the rationale for developing a new FT from TagRFP is not clear. It would be important to know what suboptimal aspect of mRubyFT would be addressed or so. Another confusing story line is the comparison to tdFTs and mFTs, which was not done in the mRubyFT paper. But what considerations are important for choosing an mFT or a tandemFT is not introduced. Subsequently, the TagFT timer development and characterization is described and compared to mRubyFT (fig 1,2, section 2.1-2.3). Then the paper describes a comparison of brightness of mTagFT, mRubyFT, mTagFT and mTsFT (fig 3), but the construction of mTsFT is only described later (in section 2.7). This is very confusing. Also the name of mTsFT is confusing since it is a tandem construct (a fusion of mTagBFP2-mScarlet).  Next to mTsFT two other tandems are described with a green and far red FP (mNeptusFT1 and mNeptusFT2) which are fusions of sfGFP to mNeptune. Also these names are misleading, In addition, the mNeptusFT1 and mNeptusFT2 were not characterized in detail in this paper. As a result, in Table I (section 2.2) 6 timer constructs appear, with one published (mRubyFT), two new TagFT timers and 3 tandem timers that are described only in section 2.7. Also the applications of the timers is presented in a way that is difficult to grasp. Instead of first testing the newly developed mTagFT sensors, a split FP version is introduced for monitoring the duration of protein-protein interactions. While this idea is nice, it is another exotic jump in the overall story line. These split-FT (fig 5) need a much more extended analysis and comparison to the non-split FT counterparts to make any conclusions with respect to their applicability. After this application the (non monomeric) TagFT timer is then applied for a gene expression timing study (Fig 6), which would be the most logic application. Yet after this application the Fucci system is introduced together with several new tandem timers (Fig 7): a mix with blue to red and green to far red timers. These tandem timers are not characterized and here it is almost impossible to not get completely lost in this section. To my surprise, the paper then continues with a structural analysis of the mTagFT timer and comparison to mTagRFP and mKate structures including a mutagenesis analysis of mTagFT. This overall story line is very confusing and after reading it all, the main incentive or key message of this paper remains unclear. 

2)    The screening for selecting the TagFT timer is not clearly describing what aspects were optimal in the selected clone and  what was actually screened for (page 3).

3)    It is well known that TagRFP is a dimer. So attempts were described to monomerize TagFT. They obtained different results with different methods (page 6, paragraph 3). In fig 2 the TagFT protein is analyzed which clearly does not elute as monomer (fig 2e), and in supplemental fig3, mTagFT is analyzed, but also here in fig S3e a substantial portion does elute at the wrong size (dimer). In this respect it is better to perform the OSER test (Costantini et al) in which TagRFP can be shown to form whorls, which very easily discriminated from monomeric RFPs like mScarlet or mCherry. So this assay should be used to compare also TagFT and mTagFT to address the conflicting data and create consensus around the monomeric or partially dimeric state of these FPs in cells. This is important especially for addressing protein-protein interactions (like the applications shown in fig 5).

4)    The blue form of TagFT and mTagFT (Fig 2d, Fig S3d) seems inferior to mRubyFT in terms of pH stability. This may be problematic in the physiological pH range where timing effects then can be obscured by pH changes leading to problems in quantification.

5)    The photobleaching curves of mTagFT and TagFT look odd (Fig S3f and 2f): they first increase and subsequently decrease. This behavior looks similar to the photobleaching curve of TagRFP-T and reflects photochromism likely involving cis-trans isomerization and bleaching happening simultaneously.

6)    TagFT and mTagFT can be photoconverted upon illumination with violet light: the blue fluorescence decreases and the red fluorescence increases. This is a highly undesirable property, which can seriously interfere with quantification of timing red/blue ratios. In addition to photoconversion, the red form of mTagFT and TagFT may also be prone to photochromism. This is not tested here, but should be tested by repeated alternating blue and green excitation and looking at the red fluorescence intensity. This may further complicate quantitative applications of these timers.

7)    The mTagFT timer was used as fusion tag (Fig 4). But these images are very suboptimal. I do not see a clear actin labeling and for alpha-tubulin no microtubule network can be seen in the images. In the blue form even some aggregates seem present. I think these images would argue against the monomeric nature of mTagFT.

8)    The split-FT experiment is conceptually appealing, but as presented here is too preliminary. It is nice that in principle a split version can be made and that this becomes fluorescent. But the resulting combined FT should have been characterized better in terms of efficiency of complementation and comparative brightness to the corresponding non-split version. In addition, in this experiment, the timing of the interaction is not controlled. In essence, this experiment is mix of a timing of the production/accumulation of both split parts (gene expression timing) and the timing of the formation of the heterodimer. The latter is probably constitutive, so what we then see is the timing of the gene expression and not of the dimerization. To do this well, one would need a system where production of the split parts and the heterodimerization can be separately controlled experimentally. This would need a completely new approach with an inducible interaction, but as such could much better sell this conceptually appealing idea. In its current form it is too preliminary.

9)    The main-stream application of fluorescent timers is the timing of gene expression, which is summarized in figure 6. But this experiment is n=1 and is very preliminary. It does not show how the TagFT timer performs in comparison to other FTs. In addition, the images show other cells with different colors at the same time point, so it is very unclear which cell then should be selected for the curves shown in fig 6c and which cells not.

10) In figure 7 the diverse timers were quantified when fused to hCdt or hGeminin. Yet it is unclear how the diverse blue and red intensities were quantified (fig a-d). It seems from the figure to be just intensity-imaging and hence it would be dependent on expression levels and all sort of microscope settings, which would be in contrast to the P2A system described in fig 3.

11) With this double timer Fucci system 4 spectrally different FP signals are produced: blue, green, red and far red. It would be expected that not all these signals can be completely spectrally separated using the spinning disk microscope system. It would help in this respect to have a description of all the excitation/emission conditions used next to the spectra of the FT constructs and a table with the normalized throughput of the 4 spectral species in the 4 detected channels. This is important if this system would be adopted in other labs with other imaging systems.

12) The crystal structure description is quite disappointing. The phenolate moiety of the chromophore of mTagFT cannot be identified (see page 17). Yet this position is absolutely crucial for the interpretation of all spectroscopic data. Then according to spectral analysis certain assumptions were made with respect to cis or trans orientation. While these assumptions may have certain validity, it is puzzling that no orientation can be concluded from the X-ray structure, and I find the presentation of the red chromophore lacking the phenolate moiety in Fig 8a-c and 9a really odd-looking and highly confusing. Even statements about hydrogen bonds to residues surrounding this phenolate moiety are made, but they cannot be substantiated by the structural data.

13) Also confusing is the statement on page 19 that “mTagFT seems to be a monomer (or weak dimer) in a crystal”. But what is it? This difference is quite crucial for biological applications (see comment 4).

14) While the TagFT and mTagFT timers show quite some differences with the TagRFP ancestor, the other timers shown incorporate hardly any differences to published FPs. For instance the mNeptusFT1 is identical to mNeptus and just a fusion of mNeptune to sfGFP (Fig S8) and the difference between mNeptusFT1 and mNeptus FT2 is a single aspartate to asparagine mutation. Furthermore, mTS is a straight fusion of mTagBFP2 to mScarlet and mTsFT contains a single lysine to arginine point mutation (Fig S9). I find the nomenclature confusing. They are tandem constructs, so their names should not start with m but with t. One could think calling them tNeptune-sfGFP-FT or tTagBFP-mScarlet-FT.

In conclusion I would strongly suggest reorganization of this manuscript. It should be split in parts. For instance, a separate publication describing TagFT and mTagFT, including comparison to existing single FP based timers (including mRubyFT) and a structural analysis (with better crystals also revealing the phenolate moiety of the chromophore). The buildup of that manuscript would be essentially similar as ref 13 and should also include an OSER analysis and address points 2-7, 9, 12 and 13). One could expand that with the application shown in fig 6, but then with much better images, true quantification and statistics. Another manuscript could deal with the tandem timers and the Fucci system. Ideally that manuscript would much better characterize and compare the different tandem timer constructs and work towards the application in the Fucci cell cycle system. There the added information on the timing of G1 and  S/G2 should ideally be better presented in a biological context as compared to the existing non-color changing Fucci system (and address comments 10, 11 and 13). The Split-timer application is too preliminary in its current form. The idea is nice, but to show that such a system can indeed provide quantitative information on the timing of a protein-protein interaction would require a controllable protein-protein interaction system as proof of concept. In addition, the properties of the split-FT after complementation should be better characterized.

Reviewer 2 Report

Reviewer Comments to Author:

This paper developed a series of true and tandem fluorescent timers (tFTs and tdFTs) with different photoconversion properties including new blue-to-red TagFT and mTagFT, optimized green-to-far-red mNeptusFT as well as blue-to-red mTsFT. Among them, optical properties of TagFT and mTagFT are well characterized in Hela and KEK293T cells. These new tools provide more possibilities to study the elaborate processes of intracellular events. For example, the FucciFT2 system based on these new FTs, could be utilized for visualizing transitions between the cell cycle. I will support this manuscript to be accepted once the following issues are fully addressed.

1)     On page 3, line 16, how is the overlap mutagenesis done? Does it mean combination mutagenesis at different sites simultaneously or just directly combine candidates with improved performance at different sites together? It will be good if the authors could clearly describe the optimization processes.

2)     Figure 2f, in the photostability test, why the red form fluorescence of the TagFT shows slightly decrease at the beginning and then increases to the peak at 250s before the continuous decrease until light illumination stopped? Similar results could also be observed in the photostability curve of red form of mTagFT (Figure S3f). It seems to have 2 phases of TagFT and mTagFT during light illumination and need to be calculated separately. Besides that, the red form of mTagFT has about 2-fold higher photostability than TagFT. Whether the mutations introduced during mTagFT development also affect the photostability and to some extent could be elucidated according to crystal structure information? mTsFT has higher brightness than TagFT and mTagFT. What about the photostability of mTsFT? The authors should clarify or discuss this point.

3)     Figure 5b, it’s better to show the example images of negative control groups that expressing bJun-TagFTN or bFos-TagFTC alone compared to the bJun-TagFTN+ bFos-TagFTC group.

4)     Figure 6, the images show a 46h-recording, but the x axis of example data just covers 0-32h. Does the red fluorescence reach the peak at 32h or further increase and then stabilize for a while? Prolonged data needs to be shown and consists with the images.

5)     Figure 7, for the application of FucciFT2 system in visualizing transitions between the cell cycle, for example, the mTsFT-hGeminin is described to preserve normal cells division. Moreover, the cells expressing FucciFT1 system are described to grow slowly and the cell cycle is elongated. However, there is no statistical analysis of different phases in the cell cycle to support these conclusions.

Minor points:

1)     The font size, color, position and length of scale bar should be unified. For example, in Figure 3 and in Figure 4, the numbers of scale bar are not clearly labeled. The authors should go through the Figures (including the supplement Figures) to clearly label all the items.

2)     On page 20, the title may be “2.9. Directed Mutagenesis of the mTagFT Timer at Key Positions”, but not the previous “2.6.”.

3)     On page 24, conclusions, line 2, “we developed and characterized a set of the novel true blue-to-red fluorescent timers, TagFT and mTagFT, derived from TagRFP”, the name of TagRFP should be unified, but not the previous “TagFT RFP”.

4)     Figure S10. Fix the legend “Visualization of the cell cycle using FucciFT1 system……”.

5)     Figure S6a, what are the differences between the blue bars and red bars? Label “433 nm” on the top is confusing.

Reviewer 3 Report

The authors developed novel fluorescent timers, characterized their photophysical properties in vitro and in HeLa cells and demonstrated the various biological applications. A nice video is presented in the Suppl. Mater. The crystal structure of one of the tFTP-s was analyzed. These are valuable results of a comprehensive study made in a hot field of fluorescence imaging. I recommend this paper for publication in IJMS. It is carefully written, I found only a few technical errors.

p. 2. ‘Stock’s shift’ correctly ‘Stokes shift

p. 6. ‘both TagFT and mTagFT’ delete ’both’

p. 7. it is not mentioned in the caption of Fig. 2c that the black trace shows the red-to-blue ratio; 550/25BP is a filter?

p. 14 ’visualizatio’

p. 24 ‘a split version of the splitTagFT timer’ probably ‘a split version of the TagFT timer’

References The numbers are duplicated

Round 2

Reviewer 1 Report

Subach et al provided answers to all my question in a detailed response. 

However, I find almost all their answers highly unsatisfactory and/or strongly underlining the diverse shortcomings of this paper.

I wrote a large piece in my first review to address the overall organization and key message of the paper that I found confusing and I made several suggestions to make a better/more logical story line to help the authors. In the first version I advised rejection because the comments are too large to be answered with some minor line editing of the story. Yet in their response 1.1-1.8 only small text additions were made, or small (defensive) explanations were given. So the overall organization of the paper remains unchanged. In 1.9 this is also explicitly stated by the authors. Yet they make the mistake that the poor organization of the paper is not my problem or shortcoming: it is their problem and it is problematic for the readers of IJMS that the organization of the paper remains a mess. 

With respect to the scientific content: the response to point 2 is adequate. 

But all other responses to the main issues underline serious weaknesses of these new probes:

The response to point 3 is not adequate: no OSER test is included (this was strongly advised), and it is admitted that dimerization is a problem: “fusion with cytoskeleton … not appropriate for these fusions”.  Yet the ‘conflicting data’ remain present regarding monomeric/dimeric nature of the protein.

The large issues in point 4-8 are a admitted ! I.e. a worse pH response (4), problems with the photobleaching data (5), problematic photochroism with no new requested experiments (6), very low quality of images in fig 4 with statement that the data “argue against the monomeric nature” (7), only copied a sentence of my remark demonstrating they did not characterize the split FP timer (8). I find these reponses very disappointing. 

The response to issue 9 is unclear. The image with many different colors remains and how cells were selected is unclear. So four cells in two cultures were selected? How? And how representative are they? For example, what percentage of cells shows the same results?

Point 10 is admitted: it is just intensity-based quantification, which is known to be very dependent on many issues.

Point 11: no extra info is given on bleed-through factors.

Point 12: the problem of the poor quality of the Xray data is admitted. I did not propose modelling. But why should one burden the readership of IJMS with poor quality X-ray data where half of the chromophore is not visible?

Point 13: no further clarity is given about the dimerization.

Point 14: admitted but no change in nomenclature is done.

Point 15: Unsatisfactory answer. Of course, the authors and not reviewers decide on how to present/organize a paper, but when they do not change the confusing story line/overall organization of the paper and admit several severe problems (4,5,6,7,8,10,12, 14) they should not be surprised that my judgement of the revised paper is even worse than their original paper.
